# RETHINKING THE VALUE OF NETWORK PRUNING

**Zhuang Liu**[1][*]**, Mingjie Sun**[2][*][†]**, Tinghui Zhou**[1]**, Gao Huang**[2]**, Trevor Darrell**[1]
[1]University of California, Berkeley   [2]Tsinghua University

## ABSTRACT

Network pruning is widely used for reducing the heavy inference cost of deep models in low-resource settings. A typical pruning algorithm is a three-stage pipeline, i.e., training (a large model), pruning and fine-tuning. During pruning, according to a certain criterion, redundant weights are pruned and important weights are kept to best preserve the accuracy. In this work, we make several surprising observations which contradict common beliefs. For all state-of-the-art structured pruning algorithms we examined, fine-tuning a pruned model only gives comparable or worse performance than training that model with randomly initialized weights. For pruning algorithms which assume a predefined target network architecture, one can get rid of the full pipeline and directly train the target network from scratch. Our observations are consistent for multiple network architectures, datasets, and tasks, which imply that: 1) training a large, over-parameterized model is often not necessary to obtain an efficient final model, 2) learned "important" weights of the large model are typically not useful for the small pruned model, 3) the pruned architecture itself, rather than a set of inherited "important" weights, is more crucial to the efficiency in the final model, which suggests that in some cases pruning can be useful as an architecture search paradigm. Our results suggest the need for more careful baseline evaluations in future research on structured pruning methods. We also compare with the "Lottery Ticket Hypothesis" (Frankle & Carbin, 2019), and find that with optimal learning rate, the "winning ticket" initialization as used in Frankle & Carbin (2019) does not bring improvement over random initialization.

## 1 INTRODUCTION

Over-parameterization is a widely-recognized property of deep neural networks (Denton et al., 2014; Ba & Caruana, 2014), which leads to high computational cost and high memory footprint for inference. As a remedy, *network pruning* (LeCun et al., 1990; Hassibi & Stork, 1993; Han et al., 2015; Molchanov et al., 2016; Li et al., 2017) has been identified as an effective technique to improve the efficiency of deep networks for applications with limited computational budget. A typical procedure of network pruning consists of three stages: 1) train a large, over-parameterized model (sometimes there are pretrained models available), 2) prune the trained large model according to a certain criterion, and 3) fine-tune the pruned model to regain the lost performance.

Generally, there are two common beliefs behind this pruning procedure. First, it is believed that starting with training a large, over-parameterized network is important (Luo et al., 2017; Carreira-Perpinán & Idelbayev, 2018), as it provides a high-performance model (due to stronger representation & optimization power) from which one can safely remove a set of redundant parameters without significantly hurting the accuracy. Therefore, this is usually believed, and reported to be superior to directly training a smaller network from scratch (Li et al., 2017; Luo et al., 2017; He et al., 2017b; Yu et al., 2018) – a commonly used baseline approach. Second, both the pruned architecture *and* its associated weights are believed to be essential for obtaining the final efficient model (Han et al.,

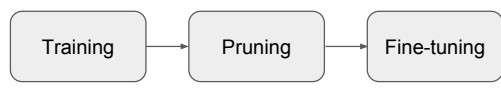

**Figure 1:** A typical three-stage network pruning pipeline.

---

[*]Equal contribution.
[†]Work done while visiting UC Berkeley.

2015). Thus most existing pruning techniques choose to *fine-tune* a pruned model instead of training it from scratch. The preserved weights after pruning are usually considered to be critical, as how to accurately select the set of important weights is a very active research topic in the literature (Molchanov et al., 2016; Li et al., 2017; Luo et al., 2017; He et al., 2017b; Liu et al., 2017; Suau et al., 2018).

In this work, we show that both of the beliefs mentioned above are not necessarily true for *structured* pruning methods, which prune at the levels of convolution channels or larger. Based on an extensive empirical evaluation of state-of-the-art pruning algorithms on multiple datasets with multiple network architectures, we make two surprising observations. First, for structured pruning methods with predefined target network architectures (Figure 2), directly training the small target model from random initialization can achieve the same, if not better, performance, as the model obtained from the three-stage pipeline. In this case, starting with a large model is not necessary and one could instead directly train the target model from scratch. Second, for structured pruning methods with auto-discovered target networks, training the pruned model from scratch can also achieve comparable or even better performance than fine-tuning. This observation shows that for these pruning methods, what matters more may be the obtained architecture, instead of the preserved weights, despite training the large model is needed to find that target architecture. Interestingly, for a *unstructured* pruning method (Han et al., 2015) that prunes individual parameters, we found that training from scratch can mostly achieve comparable accuracy with pruning and fine-tuning on smaller-scale datasets, but fails to do so on the large-scale ImageNet benchmark. Note that in some cases, if a pretrained large model is already available, pruning and fine-tuning from it can save the training time required to obtain the efficient model. The contradiction between some of our results and those reported in the literature might be explained by less carefully chosen hyper-parameters, data augmentation schemes and unfair computation budget for evaluating baseline approaches.

Our results advocate a rethinking of existing structured network pruning algorithms. It seems that the over-parameterization during the first-stage training is not as beneficial as previously thought. Also, inheriting weights from a large model is not necessarily optimal, and might trap the pruned model into a bad local minimum, even if the weights are considered "important" by the pruning criterion. Instead, our results suggest that the value of automatic structured pruning algorithms sometimes lie in identifying efficient structures and performing implicit architecture search, rather than selecting "important" weights. For most structured pruning methods which prune channels/filters, this corresponds to searching the number of channels in each layer. In section 5, we discuss this viewpoint through carefully designed experiments, and show the patterns in the pruned model could provide design guidelines for efficient architectures.

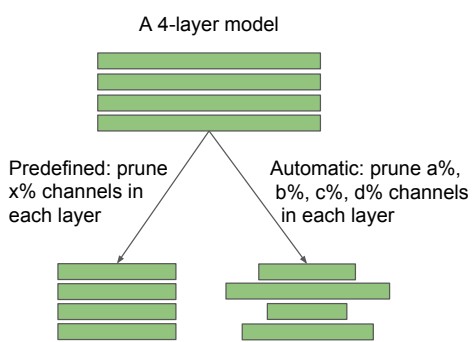

**Figure 2:** Difference between predefined and automatically discovered target architectures, in channel pruning as an example. The pruning ratio $x$ is user-specified, while $a, b, c, d$ are determined by the pruning algorithm. Unstructured sparse pruning can also be viewed as automatic.

The rest of the paper is organized as follows: in Section 2, we introduce background and some related works on network pruning; in Section 3, we describe our methodology for training the pruned model from scratch; in Section 4 we experiment on various pruning methods and show our main results for both pruning methods with predefined or automatically discovered target architectures; in Section 5, we discuss the value of automatic pruning methods in searching efficient network architectures; in Section 6 we discuss some implications and conclude the paper.

## 2 BACKGROUND

Recent success of deep convolutional networks (LeCun et al., 1998; Deng et al., 2009; Girshick et al., 2014; Long et al., 2015; He et al., 2016; 2017a) has been coupled with increased requirement of computation resources. In particular, the model size, memory footprint, the number of computation operations (FLOPs) and power usage are major aspects inhibiting the use of deep neural networks

in some resource-constrained settings. Those large models can be infeasible to store, and run in real time on embedded systems. To address this issue, many methods have been proposed such as low-rank approximation of weights (Denton et al., 2014; Lebedev et al., 2014), weight quantization (Courbariaux et al., 2016; Rastegari et al., 2016), knowledge distillation (Hinton et al., 2014; Romero et al., 2015) and network pruning (Han et al., 2015; Li et al., 2017), among which network pruning has gained notable attention due to their competitive performance and compatibility.

One major branch of network pruning methods is individual weight pruning, and it dates back to Optimal Brain Damage (LeCun et al., 1990) and Optimal Brain Surgeon (Hassibi & Stork, 1993), which prune weights based on Hessian of the loss function. More recently, Han et al. (2015) proposes to prune network weights with small magnitude, and this technique is further incorporated into the "Deep Compression" pipeline (Han et al., 2016b) to obtain highly compressed models. Srinivas & Babu (2015) proposes a data-free algorithm to remove redundant neurons iteratively. Molchanov et al. (2017) uses Variatonal Dropout (P. Kingma et al., 2015) to prune redundant weights. Louizos et al. (2018) learns sparse networks through $L_0$-norm regularization based on stochastic gate. However, one drawback of these *unstructured* pruning methods is that the resulting weight matrices are sparse, which cannot lead to compression and speedup without dedicated hardware/libraries (Han et al., 2016a).

In contrast, *structured* pruning methods prune at the level of channels or even layers. Since the original convolution structure is still preserved, no dedicated hardware/libraries are required to realize the benefits. Among structured pruning methods, channel pruning is the most popular, since it operates at the most fine-grained level while still fitting in conventional deep learning frameworks. Some heuristic methods include pruning channels based on their corresponding filter weight norm (Li et al., 2017) and average percentage of zeros in the output (Hu et al., 2016). Group sparsity is also widely used to smooth the pruning process after training (Wen et al., 2016; Alvarez & Salzmann, 2016; Lebedev & Lempitsky, 2016; Zhou et al., 2016). Liu et al. (2017) and Ye et al. (2018) impose sparsity constraints on channel-wise scaling factors during training, whose magnitudes are then used for channel pruning. Huang & Wang (2018) uses a similar technique to prune coarser structures such as residual blocks. He et al. (2017b) and Luo et al. (2017) minimizes next layer's feature reconstruction error to determine which channels to keep. Similarly, Yu et al. (2018) optimizes the reconstruction error of the final response layer and propagates a "importance score" for each channel. Molchanov et al. (2016) uses Taylor expansion to approximate each channel's influence over the final loss and prune accordingly. Suau et al. (2018) analyzes the intrinsic correlation within each layer and prune redundant channels. Chin et al. (2018) proposes a layer-wise compensate filter pruning algorithm to improve commonly-adopted heuristic pruning metrics. He et al. (2018a) proposes to allow pruned filters to recover during the training process. Lin et al. (2017); Wang et al. (2017) prune certain structures in the network based on the current input.

Our work is also related to some recent studies on the characteristics of pruning algorithms. Mittal et al. (2018) shows that random channel pruning (Anwar & Sung, 2016) can perform on par with a variety of more sophisticated pruning criteria, demonstrating the plasticity of network models. In the context of unstructured pruning, The Lottery Ticket Hypothesis (Frankle & Carbin, 2019) conjectures that certain connections together with their randomly initialized weights, can enable a comparable accuracy with the original network when trained in isolation. We provide comparisons between Frankle & Carbin (2019) and this work in Appendix A. Zhu & Gupta (2018) shows that training a small-dense model cannot achieve the same accuracy as a pruned large-sparse model with identical memory footprint. In this work, we reveal a different and rather surprising characteristic of structured network pruning methods: fine-tuning the pruned model with inherited weights is not better than training it from scratch; the resulting pruned architectures are more likely to be what brings the benefit.

## 3 METHODOLOGY

In this section, we describe in detail our methodology for training a small target model from scratch.

**Target Pruned Architectures.** We first divide network pruning methods into two categories. In a pruning pipeline, the target pruned model's architecture can be determined by either a human (i.e., predefined) or the pruning algorithm (i.e., automatic) (see Figure 2).

When a human predefines the target architecture, a common criterion is the ratio of channels to prune in each layer. For example, we may want to prune 50% channels in each layer of VGG. In this

case, no matter which specific channels are pruned, the pruned target architecture remains the same, because the pruning algorithm only *locally* prunes the least important 50% channels in each layer. In practice, the ratio in each layer is usually selected through empirical studies or heuristics. Examples of predefined structured pruning include Li et al. (2017), Luo et al. (2017), He et al. (2017b) and He et al. (2018a)

When the target architecture is automatically determined by a pruning algorithm, it is usually based on a pruning criterion that *globally* compares the importance of structures (e.g., channels) across layers. Examples of automatic structured pruning include Liu et al. (2017), Huang & Wang (2018), Molchanov et al. (2016) and Suau et al. (2018).

Unstructured pruning (Han et al., 2015; Molchanov et al., 2017; Louizos et al., 2018) also falls in the category of automatic methods, where the positions of pruned weights are determined by the training process and the pruning algorithm, and it is usually not possible to predefine the positions of zeros before training starts.

**Datasets, Network Architectures and Pruning Methods.** In the network pruning literature, CIFAR-10, CIFAR-100 (Krizhevsky, 2009), and ImageNet (Deng et al., 2009) datasets are the de-facto benchmarks, while VGG (Simonyan & Zisserman, 2015), ResNet (He et al., 2016) and DenseNet (Huang et al., 2017) are the common network architectures. We evaluate four predefined pruning methods, Li et al. (2017), Luo et al. (2017), He et al. (2017b), He et al. (2018a), two automatic structured pruning methods, Liu et al. (2017), Huang & Wang (2018), and one unstructured pruning method (Han et al., 2015). For the first six methods, we evaluate using the same (target model, dataset) pairs as presented in the original paper to keep our results comparable. For the last one (Han et al., 2015), we use the aforementioned architectures instead, since the ones in the original paper are no longer state-of-the-art. On CIFAR datasets, we run each experiment with 5 random seeds, and report the mean and standard deviation of the accuracy.

**Training Budget.** One crucial question is how long we should train the small pruned model from scratch. Naively training for the same number of epochs as we train the large model might be unfair, since the small pruned model requires significantly less computation for one epoch. Alternatively, we could compute the floating point operations (FLOPs) for both the pruned and large models, and choose the number of training epoch for the pruned model that would lead to the same amount of computation as training the large model. Note that it is not clear how to train the models to "full convergence" given the stepwise decaying learning rate schedule commonly used in the CIFAR/ImageNet classification tasks.

In our experiments, we use **Scratch-E** to denote training the small pruned models for the same epochs, and **Scratch-B** to denote training for the same amount of computation budget (on ImageNet, if the pruned model saves more than $2\times$ FLOPs, we just double the number of epochs for training Scratch-B, which amounts to less computation budget than large model training). When extending the number of epochs in Scratch-B, we also extend the learning rate decay schedules proportionally. One may argue that we should instead train the small target model for fewer epochs since it may converge faster. However, in practice we found that increasing the training epochs within a reasonable range is rarely harmful. In our experiments we found in most times Scratch-E is enough while in other cases Scratch-B is needed for a comparable accuracy as fine-tuning. Note that our evaluations use the same computation as large model training without considering the computation in fine-tuning, since in our evaluated methods fine-tuning does not take too long; if anything this still favors the pruning and fine-tuning pipeline.

**Implementation.** In order to keep our setup as close to the original papers as possible, we use the following protocols: 1) ff a previous pruning method's training setup is publicly available, e.g. Liu et al. (2017), Huang & Wang (2018) and He et al. (2018a), we adopt the original implementation; 2) otherwise, for simpler pruning methods, e.g., Li et al. (2017) and Han et al. (2015), we re-implement the three-stage pruning procedure and generally achieve similar results as in the original papers; 3) for the remaining two methods (Luo et al., 2017; He et al., 2017b), the pruned models are publicly available but without the training setup, thus we choose to re-train both large and small target models from scratch. Interestingly, the accuracy of our re-trained large model is higher than what is reported in the original papers. This could be due to the difference in the deep learning frameworks: we used Pytorch (Paszke et al., 2017) while the original papers used Caffe (Jia et al., 2014). In this case, to

accommodate the effects of different frameworks and training setups, we report the relative accuracy drop from the unpruned large model.

We use standard training hyper-parameters and data-augmentation schemes, which are used both in standard image classification models (He et al., 2016; Huang et al., 2017) and network pruning methods (Li et al., 2017; Liu et al., 2017; Huang & Wang, 2018; He et al., 2018a). The optimization method is SGD with Nesterov momentum, using an stepwise decay learning rate schedule. For random weight initialization, we adopt the scheme proposed in (He et al., 2015). For results of models fine-tuned from inherited weights, we either use the released models from original papers (case 3 above) or follow the common practice of fine-tuning the model using the lowest learning rate when training the large model (Li et al., 2017; He et al., 2017b). For CIFAR, training/fine-tuning takes 160/40 epochs. For ImageNet, training/fine-tuning takes 90/20 epochs. For reproducing the results and a more detailed knowledge about the settings, see our code at: `https://github.com/Eric-mingjie/rethinking-network-pruning`.

## 4 EXPERIMENTS

In this section we present our experimental results comparing training pruned models from scratch and fine-tuning from inherited weights, for both predefined and automatic (Figure 2) structured pruning, as well as a magnitude-based unstructured pruning method (Han et al., 2015). We also include a comparison with the Lottery Ticket Hypothesis (Frankle & Carbin, 2019), and an experiment on transfer learning from image classification to object detection in Appendix, due to space limit. We also put the results and discussions on a pruning method (Soft Filter pruning (He et al., 2018a)) in Appendix.

### 4.1 PREDEFINED STRUCTURED PRUNING

$L_1$**-norm based Filter Pruning (Li et al., 2017)** is one of the earliest works on filter/channel pruning for convolutional networks. In each layer, a certain percentage of filters with smaller $L_1$-norm will be pruned. Table 1 shows our results. The Pruned Model column shows the list of predefined target models (see (Li et al., 2017) for configuration details on each model). We observe that in each row, scratch-trained models achieve at least the same level of accuracy as fine-tuned models, with Scratch-B slightly higher than Scratch-E in most cases. On ImageNet, both Scratch-B models are better than the fine-tuned ones by a noticeable margin.

| Dataset | Model | Unpruned | Pruned Model | Fine-tuned | Scratch-E | Scratch-B |
|---------|-------|----------|--------------|------------|-----------|-----------|
| CIFAR-10 | VGG-16 | 93.63 ($\pm$0.16) | VGG-16-A | 93.41 ($\pm$0.12) | 93.62 ($\pm$0.11) | **93.78** ($\pm$0.15) |
| | ResNet-56 | 93.14 ($\pm$0.12) | ResNet-56-A | 92.97 ($\pm$0.17) | 92.96 ($\pm$0.26) | **93.09** ($\pm$0.14) |
| | | | ResNet-56-B | 92.67 ($\pm$0.14) | 92.54 ($\pm$0.19) | **93.05** ($\pm$0.18) |
| | ResNet-110 | 93.14 ($\pm$0.24) | ResNet-110-A | 93.14 ($\pm$0.16) | **93.25** ($\pm$0.29) | 93.22 ($\pm$0.22) |
| | | | ResNet-110-B | 92.69 ($\pm$0.09) | 92.89 ($\pm$0.43) | **93.60** ($\pm$0.25) |
| ImageNet | ResNet-34 | 73.31 | ResNet-34-A | 72.56 | 72.77 | **73.03** |
| | | | ResNet-34-B | 72.29 | 72.55 | **72.91** |

**Table 1:** Results (accuracy) for $L_1$-norm based filter pruning (Li et al., 2017). "Pruned Model" is the model pruned from the large model. Configurations of Model and Pruned Model are both from the original paper.

**ThiNet (Luo et al., 2017)** greedily prunes the channel that has the smallest effect on the next layer's activation values. As shown in Table 2, for VGG-16 and ResNet-50, both Scratch-E and Scratch-B can almost always achieve better performance than the fine-tuned model, often by a significant margin. The only exception is Scratch-E for VGG-Tiny, where the model is pruned very aggressively from VGG-16 (FLOPs reduced by $15\times$), and as a result, drastically reducing the training budget for Scratch-E. The training budget of Scratch-B for this model is also 7 times smaller than the original large model, yet it can achieve the same level of accuracy as the fine-tuned model.

**Regression based Feature Reconstruction (He et al., 2017b)** prunes channels by minimizing the feature map reconstruction error of the next layer. In contrast to ThiNet (Luo et al., 2017), this optimization problem is solved by LASSO regression. Results are shown in Table 3. Again, in terms of relative accuracy drop from the large models, scratch-trained models are better than the fine-tuned models.

| Dataset | Unpruned | Strategy | Pruned Model | | |
|---------|----------|----------|--------------|--------------|--------------|
| ImageNet | VGG-16 | | VGG-Conv | VGG-GAP | VGG-Tiny |
| | 71.03 | Fine-tuned | $-1.23$ | $-3.67$ | $-11.61$ |
| | 71.51 | Scratch-E | $-2.75$ | $-4.66$ | $-14.36$ |
| | | Scratch-B | $+\mathbf{0.21}$ | $-\mathbf{2.85}$ | $-\mathbf{11.58}$ |
| | ResNet-50 | | ResNet50-30% | ResNet50-50% | ResNet50-70% |
| | 75.15 | Fine-tuned | $-6.72$ | $-4.13$ | $-3.10$ |
| | 76.13 | Scratch-E | $-5.21$ | $-2.82$ | $-1.71$ |
| | | Scratch-B | $-\mathbf{4.56}$ | $-\mathbf{2.23}$ | $-\mathbf{1.01}$ |

**Table 2:** Results (accuracy) for ThiNet (Luo et al., 2017). Names such as "VGG-GAP" and "ResNet50-30%" are pruned models whose configurations are defined in Luo et al. (2017). To accommodate the effects of different frameworks between our implementation and the original paper's, we compare relative accuracy drop from the unpruned large model. For example, for the pruned model VGG-Conv, $-1.23$ is relative to 71.03 on the left, which is the reported accuracy of the unpruned large model VGG-16 in the original paper; $-2.75$ is relative to 71.51 on the left, which is VGG-16's accuracy in our implementation.

| Dataset | Unpruned | Strategy | Pruned Model |
|---------|----------|----------|--------------|
| ImageNet | VGG-16 | | VGG-16-5x |
| | 71.03 | Fine-tuned | $-2.67$ |
| | 71.51 | Scratch-E | $-3.46$ |
| | | Scratch-B | $-\mathbf{0.51}$ |
| | ResNet-50 | | ResNet-50-2x |
| | 75.51 | Fine-tuned | $-3.25$ |
| | 76.13 | Scratch-E | $-1.55$ |
| | | Scratch-B | $-\mathbf{1.07}$ |

**Table 3:** Results (accuracy) for Regression based Feature Reconstruction (He et al., 2017b). Pruned models such as "VGG-16-5x" are defined in He et al. (2017b). Similar to Table 2, we compare relative accuracy drop from unpruned large models.

## 4.2 AUTOMATIC STRUCTURED PRUNING

**Network Slimming (Liu et al., 2017)** imposes $L_1$-sparsity on channel-wise scaling factors from Batch Normalization layers (Ioffe & Szegedy, 2015) during training, and prunes channels with lower scaling factors afterward. Since the channel scaling factors are compared across layers, this method produces automatically discovered target architectures. As shown in Table 4, for all networks, the small models trained from scratch can reach the same accuracy as the fine-tuned models. More specifically, we found that Scratch-B consistently outperforms (8 out of 10 experiments) the fine-tuned models, while Scratch-E is slightly worse but still mostly within the standard deviation.

| Dataset | Model | Unpruned | Prune Ratio | Fine-tuned | Scratch-E | Scratch-B |
|---------|-------|----------|-------------|------------|-----------|-----------|
| CIFAR-10 | VGG-19 | 93.53 ($\pm$0.16) | 70% | 93.60 ($\pm$0.16) | 93.30 ($\pm$0.11) | **93.81** ($\pm$0.14) |
| | PreResNet-164 | 95.04 ($\pm$0.16) | 40% | 94.77 ($\pm$0.12) | 94.70 ($\pm$0.11) | **94.90** ($\pm$0.04) |
| | | | 60% | 94.23 ($\pm$0.21) | 94.58 ($\pm$0.18) | **94.71** ($\pm$0.21) |
| | DenseNet-40 | 94.10 ($\pm$0.12) | 40% | 94.00 ($\pm$0.20) | 93.68 ($\pm$0.18) | **94.06** ($\pm$0.12) |
| | | | 60% | **93.87** ($\pm$0.13) | 93.58 ($\pm$0.21) | 93.85 ($\pm$0.25) |
| CIFAR-100 | VGG-19 | 72.63 ($\pm$0.21) | 50% | 72.32 ($\pm$0.28) | 71.94 ($\pm$0.17) | **73.08** ($\pm$0.22) |
| | PreResNet-164 | 76.80 ($\pm$0.19) | 40% | 76.22 ($\pm$0.20) | 76.36 ($\pm$0.32) | **76.68** ($\pm$0.35) |
| | | | 60% | 74.17 ($\pm$0.33) | 75.05 ($\pm$ 0.08) | **75.73** ($\pm$0.29) |
| | DenseNet-40 | 73.82 ($\pm$0.34) | 40% | **73.35** ($\pm$0.17) | 73.24 ($\pm$0.29) | 73.19 ($\pm$0.26) |
| | | | 60% | 72.46 ($\pm$0.22) | 72.62 ($\pm$0.36) | **72.91** ($\pm$0.34) |
| ImageNet | VGG-11 | 70.84 | 50% | 68.62 | 70.00 | **71.18** |

**Table 4:** Results (accuracy) for Network Slimming (Liu et al., 2017). "Prune ratio" stands for total percentage of channels that are pruned in the whole network. The same ratios for each model are used as the original paper.

**Sparse Structure Selection (Huang & Wang, 2018)** also uses sparsified scaling factors to prune structures, and can be seen as a generalization of Network Slimming. Other than channels, pruning

can be on residual blocks in ResNet or groups in ResNeXt (Xie et al., 2017). We examine residual blocks pruning, where ResNet-50 are pruned to be ResNet-41, ResNet-32 and ResNet-26. Table 5 shows our results. On average Scratch-E outperforms pruned models, and for all models Scratch-B is better than both.

| Dataset | Model | Unpruned | Pruned Model | Pruned | Scratch-E | Scratch-B |
|---------|-------|----------|--------------|--------|-----------|-----------|
| ImageNet | ResNet-50 | 76.12 | ResNet-41 | 75.44 | 75.61 | **76.17** |
| | | | ResNet-32 | 74.18 | 73.77 | **74.67** |
| | | | ResNet-26 | 71.82 | 72.55 | **73.41** |

**Table 5:** Results (accuracy) for residual block pruning using Sparse Structure Selection (Huang & Wang, 2018). In the original paper no fine-tuning is required so there is a "Pruned" column instead of "Fine-tuned" as before.

### 4.3 UNSTRUCTURED MAGNITUDE-BASED PRUNING (HAN ET AL., 2015)

Unstructured magnitude-based weight pruning (Han et al., 2015) can also be treated as automatically discovering architectures, since the positions of exact zeros cannot be determined before training, but we highlight its differences with structured pruning using another subsection. Because all the network architectures we evaluated are fully-convolutional (except for the last fully-connected layer), for simplicity, we only prune weights in convolution layers here. Before training the pruned sparse model from scratch, we re-scale the standard deviation of the Gaussian distribution for weight initialization, based on how many non-zero weights remain in this layer. This is to keep a constant scale of backward gradient signal as in (He et al., 2015), which however in our observations does not bring gains compared with unscaled counterparts.

| Dataset | Model | Unpruned | Prune Ratio | Fine-tuned | Scratch-E | Scratch-B |
|---------|-------|----------|-------------|------------|-----------|-----------|
| CIFAR-10 | VGG-19 | 93.50 (±0.11) | 30% | 93.51 (±0.05) | **93.71** (±0.09) | 93.31 (±0.26) |
| | | | 80% | 93.52 (±0.10) | **93.71** (±0.08) | 93.64 (±0.09) |
| | | | 95% | 93.34 (±0.13) | 93.21 (±0.17) | **93.63** (±0.18) |
| | PreResNet-110 | 95.04 (±0.15) | 30% | 95.06 (±0.05) | 94.84 (±0.07) | **95.11** (±0.09) |
| | | | 80% | **94.55** (±0.11) | 93.76 (±0.10) | 94.52 (±0.13) |
| | | | 95% | **92.35** (±0.20) | 91.23 (±0.11) | 91.55 (±0.34) |
| | DenseNet-BC-100 | 95.24 (±0.17) | 30% | 95.21 (±0.17) | 95.22 (±0.18) | **95.23** (±0.14) |
| | | | 80% | 95.04 (±0.15) | 94.42 (±0.12) | **95.12** (±0.04) |
| | | | 95% | **94.19** (±0.15) | 92.91 (±0.22) | 93.44 (±0.19) |
| CIFAR-100 | VGG-19 | 71.70 (±0.31) | 30% | 71.96 (±0.36) | 72.81 (±0.31) | **73.30** (±0.25) |
| | | | 50% | 71.85 (±0.30) | 73.12 (±0.36) | **73.77** (±0.23) |
| | | | 95% | 70.22 (±0.38) | 70.88 (±0.35) | **72.08** (±0.15) |
| | PreResNet-110 | 76.96 (±0.34) | 30% | 76.88 (±0.31) | 76.36 (±0.26) | **76.96** (±0.31) |
| | | | 50% | **76.60** (±0.36) | 75.45 (±0.23) | 76.42 (±0.39) |
| | | | 95% | 68.55 (±0.51) | 68.13 (±0.64) | **68.99** (±0.32) |
| | DenseNet-BC-100 | 77.59 (±0.19) | 30% | 77.23 (±0.05) | 77.58 (±0.25) | **77.97** (±0.31) |
| | | | 50% | 77.41 (±0.14) | 77.65 (±0.09) | **77.80** (±0.23) |
| | | | 95% | **73.67** (±0.03) | 71.47 (±0.46) | 72.57 (±0.37) |
| ImageNet | VGG-16 | 73.37 | 30% | 73.68 | 72.75 | **74.02** |
| | | | 60% | **73.63** | 71.50 | 73.42 |
| | ResNet-50 | 76.15 | 30% | **76.06** | 74.77 | 75.70 |
| | | | 60% | **76.09** | 73.69 | 74.91 |

**Table 6:** Results (accuracy) for unstructured pruning (Han et al., 2015). "Prune Ratio" denotes the percentage of parameters pruned in the set of all convolutional weights.

As shown in Table 6, on the smaller-scale CIFAR datasets, when the pruned ratio is small ($\leq 80\%$), Scratch-E sometimes falls short of the fine-tuned results, but Scratch-B is able to perform at least on par with the latter. However, we observe that in some cases, when the prune ratio is large (95%), fine-tuning can outperform training from scratch. On the large-scale ImageNet dataset, we note that the Scratch-B result is mostly worse than fine-tuned result by a noticable margin, despite at a decent accuracy level. This could be due to the increased difficulty of directly training on the highly sparse networks (CIFAR), or the scale/complexity of the dataset itself (ImageNet). Another possible reason is that compared with structured pruning, unstructured pruning significantly changes

the weight distribution (more details in Appendix G). The difference in scratch-training behaviors also suggests an important difference between structured and unstructured pruning.

# 5 NETWORK PRUNING AS ARCHITECTURE SEARCH

While we have shown that, for structured pruning, the inherited weights in the pruned architecture are not better than random, the pruned architecture itself turns out to be what brings the efficiency benefits. In this section, we assess the value of architecture search for automatic network pruning algorithms (Figure 2) by comparing pruning-obtained models and uniformly pruned models. Note that the connection between network pruning and architecture learning has also been made in prior works (Han et al., 2015; Liu et al., 2017; Gordon et al., 2018; Huang et al., 2018), but to our knowledge we are the first to isolate the effect of inheriting weights and solely compare pruning-obtained architectures with uniformly pruned ones, by training both of them from scratch.

**Parameter Efficiency of Pruned Architectures.** In Figure 3(left), we compare the parameter efficiency of architectures obtained by an automatic channel pruning method (Network Slimming (Liu et al., 2017)), with a naive predefined pruning strategy that uniformly prunes the same percentage of channels in each layer. All architectures are trained from random initialization for the same number of epochs. We see that the architectures obtained by Network Slimming are more parameter efficient, as they could achieve the same level of accuracy using $5\times$ fewer parameters than uniformly pruning architectures. For unstructured magnitude-based pruning (Han et al., 2015), we conducted a similar experiment shown in Figure 3 (right). Here we uniformly sparsify all individual weights at a fixed probability, and the architectures obtained this way are much less efficient than the pruned architectures.

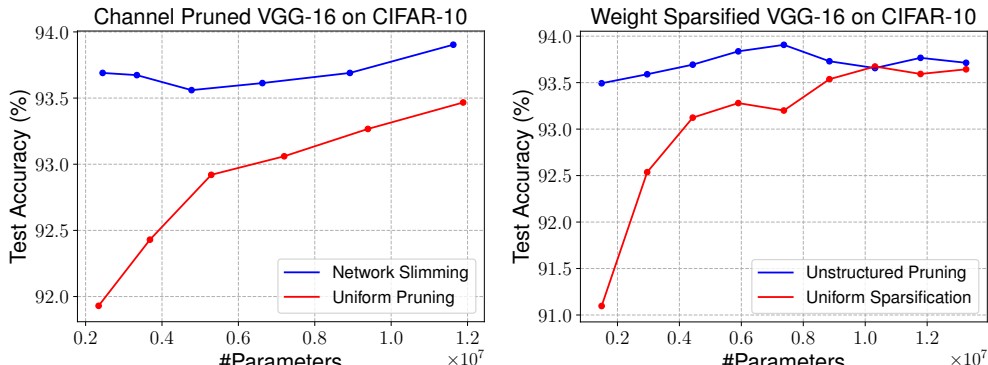

**Figure 3:** Pruned architectures obtained by different approaches, all *trained from scratch*, averaged over 5 runs. Architectures obtained by automatic pruning methods (*Left:* Network Slimming (Liu et al., 2017), *Right:* Unstructured pruning (Han et al., 2015)) have better parameter efficiency than uniformly pruning channels or sparsifying weights in the whole network.

| Layer | Width | Width* | Layer | Width | Width* |
|-------|-------|--------|-------|-------|--------|
| 1 | 64 | 39.0±3.7 | 8 | 512 | 217.3±6.6 |
| 2 | 64 | 64.0±0.0 | 9 | 512 | 120.0±4.4 |
| 3 | 128 | 127.8±0.4 | 10 | 512 | 63.0±1.9 |
| 4 | 128 | 128.0±0.0 | 11 | 512 | 47.8±2.9 |
| 5 | 256 | 255.0±1.0 | 12 | 512 | 62.0±3.4 |
| 6 | 256 | 250.5±0.5 | 13 | 512 | 88.8±3.1 |
| 7 | 256 | 226.0±2.5 | Total | 4224 | 1689.2 |

**Table 7:** Network architectures obtained by pruning 60% channels on VGG-16 (in total 13 conv-layers) using Network Slimming. Width and Width* are number of channels in the original and pruned architectures, averaged over 5 runs.

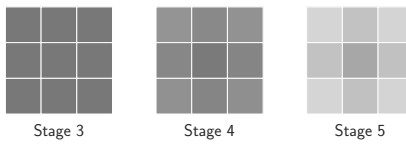

**Figure 4:** The average sparsity pattern of all $3\times3$ convolutional kernels in certain layer stages in a unstructured pruned VGG-16. Darker color means higher probability of weight being kept.

We also found the channel/weight pruned architectures exhibit very consistent patterns (see Table 7 and Figure 4). This suggests the original large models may be redundantly designed for the task and

the pruning algorithm can help us improve the efficiency. This also confirms the value of automatic pruning methods for searching efficient models on the architectures evaluated.

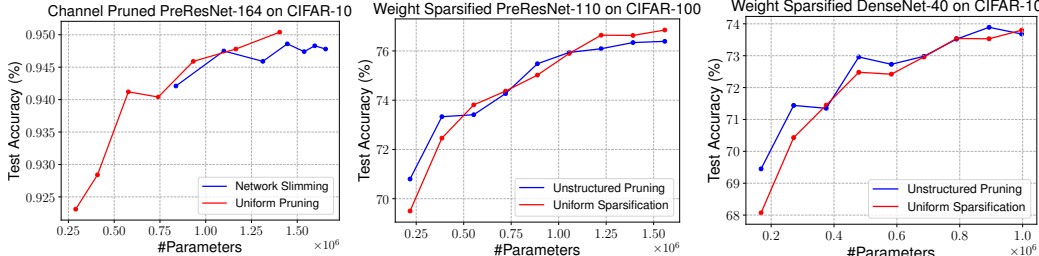

**Figure 5:** Pruned architectures obtained by different approaches, *all trained from scratch*, averaged over 5 runs. *Left:* Results for PreResNet-164 pruned on CIFAR-10 by Network Slimming (Liu et al., 2017). *Middle* and *Right*: Results for PreResNet-110 and DenseNet-40 pruned on CIFAR-100 by unstructured pruning (Han et al., 2015).

**More Analysis.** However, there also exist cases where the architectures obtained by pruning are not better than uniformly pruned ones. We present such results in Figure 5, where the architectures obtained by pruning (blue) are not significantly more efficient than uniform pruned architectures (red). This phenomenon happens more likely on modern architectures like ResNets and DenseNets. When we investigate the sparsity patterns of those pruned architectures (shown in Table 18, 19 and 20 in Appendix H), we find that they exhibit near-uniform sparsity patterns across stages, which might be the reason why it can only perform on par with uniform pruning. In contrast, for VGG, the pruned sparsity patterns can always beat the uniform ones as shown in Figure 3 and Figure 6. We also show the sparsity patterns of VGG pruned by Network Slimming (Liu et al., 2017) in Table 21 of Appendix H, and they are rather far from uniform. Compared to ResNet and DenseNet, we can see that VGG's redundancy is rather imbalanced across layer stages. Network pruning techniques may help us identify the redundancy better in the such cases.

**Generalizable Design Principles from Pruned Architectures.** Given that the automatically discovered architectures tend to be parameter efficient on the VGG networks, one may wonder: can we derive generalizable principles from them on how to design a better architecture? We conduct several experiments to answer this question.

For Network Slimming, we use the average number of channels in each layer stage (layers with the same feature map size) from pruned architectures to construct a new set of architectures, and we call this approach "Guided Pruning"; for magnitude-based pruning, we analyze the sparsity patterns (Figure 4) in the pruned architectures, and apply them to construct a new set of sparse models, which we call "Guided Sparsification". The results are shown in Figure 6. It can be seen that for both Network Slimming (Figure 6 left) and unstructured pruning (Figure 6 right), guided design of architectures (green) can perform on par with pruned architectures (blue).

Interestingly, these guided design patterns can sometimes be transferred to a different VGG-variant and/or dataset. In Figure 6, we distill the patterns of pruned architectures from VGG-16 on CIFAR-10 and apply them to design efficient VGG-19 on CIFAR-100. These sets of architectures are denoted as "Transferred Guided Pruning/Sparsification". We can observe that they (brown) may sometimes be slightly worse than architectures directly pruned (blue), but are significantly better than uniform pruning/sparsification (red). In these cases, one does not need to train a large model to obtain an efficient model as well, as transferred design patterns can help us achieve the efficiency directly.

**Discussions with Conventional Architecture Search Methods.** Popular techniques for network architecture search include reinforcement learning (Zoph & Le, 2017; Baker et al., 2017) and evolutionary algorithms (Xie & Yuille, 2017; Liu et al., 2018a). In each iteration, a randomly initialized network is trained and evaluated to guide the search, and the search process usually requires thousands of iterations to find the goal architecture. In contrast, using network pruning as architecture search only requires a one-pass training, however the search space is restricted to the set of all "subnetworks" inside a large network, whereas traditional methods can search for more variations, e.g., activation functions or different layer orders.

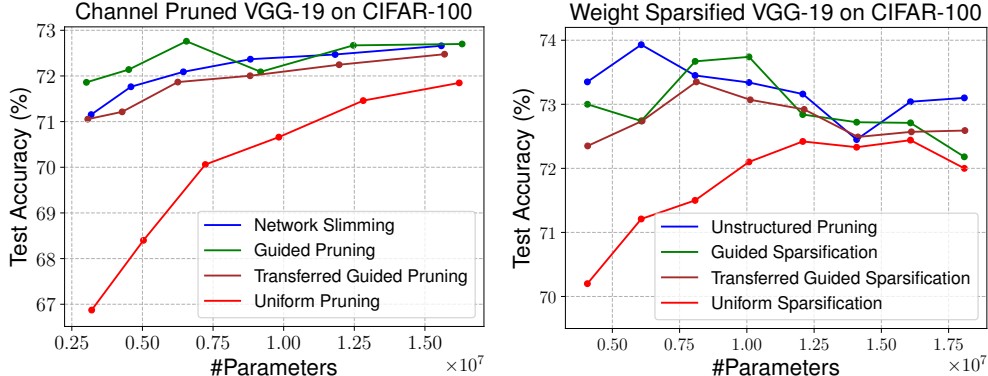

**Figure 6:** Pruned architectures obtained by different approaches, *all trained from scratch*, averaged over 5 runs. "Guided Pruning/Sparsification" means using the average sparsity patterns in each layer stage to design the network; "Transferred Guided Pruning/Sparsification" means using the sparsity patterns obtained by a pruned VGG-16 on CIFAR-10, to design the network for VGG-19 on CIFAR-100. Following the design guidelines provided by the pruned architectures, we achieve better parameter efficiency, even when the guidelines are transferred from another dataset and model.

Recently, Gordon et al. (2018) uses a similar pruning technique to Network Slimming (Liu et al., 2017) to automate the design of network architectures; He et al. (2018c) prune channels using reinforcement learning and automatically compresses the architecture. On the other hand, in the network architecture search literature, sharing/inheriting trained parameters (Pham et al., 2018; Liu et al., 2018b) during searching has become a popular approach for reducing the training budgets, but once the target architecture is found, it is still trained from scratch to maximize the accuracy.

## 6 DISCUSSION AND CONCLUSION

Our results encourage more careful and fair baseline evaluations of structured pruning methods. In addition to high accuracy, training predefined target models from scratch has the following benefits over conventional network pruning procedures: a) since the model is smaller, we can train the model using less GPU memory and possibly faster than training the original large model; b) there is no need to implement the pruning criterion and procedure, which sometimes requires fine-tuning layer by layer (Luo et al., 2017) and/or needs to be customized for different network architectures (Li et al., 2017; Liu et al., 2017); c) we avoid tuning additional hyper-parameters involved in the pruning procedure.

Our results do support the viewpoint that automatic structured pruning finds efficient architectures in some cases. However, if the accuracy of pruning and fine-tuning is achievable by training the pruned model from scratch, it is also important to evaluate the pruned architectures against uniformly pruned baselines (both training from scratch), to demonstrate the method's value in identifying efficient architectures. If the uniformly pruned models are not worse, one could also skip the pipeline and train them from scratch.

Even if pruning and fine-tuning fails to outperform the mentioned baselines in terms of accuracy, there are still some cases where using this conventional wisdom can be much faster than training from scratch: a) when a pre-trained large model is already given and little or no training budget is available; we also note that pre-trained models can only be used when the method does not require modifications to the large model training process; b) there is a need to obtain multiple models of different sizes, or one does not know what the desirable size is, in which situations one can train a large model and then prune it by different ratios.

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

# APPENDIX

## A    EXPERIMENTS ON THE LOTTERY TICKET HYPOTHESIS (FRANKLE & CARBIN, 2019)

The Lottery Ticket Hypothesis (Frankle & Carbin, 2019) conjectures that inside the large network, a sub-network together with their initialization makes the training particularly effective, and together they are termed the "winning ticket". In this hypothesis, the original initialization of the sub-network (before large model training) is needed for it to achieve competitive performance when trained in isolation. Their experiments show that training the sub-network with randomly re-initialized weights performs worse than training it with the original initialization inside the large network. In contrast, our work does not require reuse of the original initialization of the pruned model, and shows that random initialization is enough for the pruned model to achieve competitive performance.

The conclusions seem to be contradictory, but there are several important differences in the evaluation settings: a) Our main conclusion is drawn on *structured* pruning methods, despite for small-scale problems (CIFAR) it also holds on unstructured pruning; Frankle & Carbin (2019) only evaluates on unstructured pruning. b) Our evaluated network architectures are all relatively large modern models used in the original pruning methods, while most of the experiments in Frankle & Carbin (2019) use small shallow networks (< 6 layers). c) We use momentum SGD with a large initial learning rate (0.1), which is widely used in prior image classification (He et al., 2016; Huang et al., 2017) and pruning works (Li et al., 2017; Liu et al., 2017; He et al., 2017b; Luo et al., 2017; He et al., 2018a; Huang & Wang, 2018) to achieve high accuracy, and is the de facto default optimization setting on CIFAR and ImageNet; while Frankle & Carbin (2019) mostly uses Adam with much lower learning rates. d) Our experiments include the large-scale ImageNet dataset, while Frankle & Carbin (2019) only considers MNIST and CIFAR.

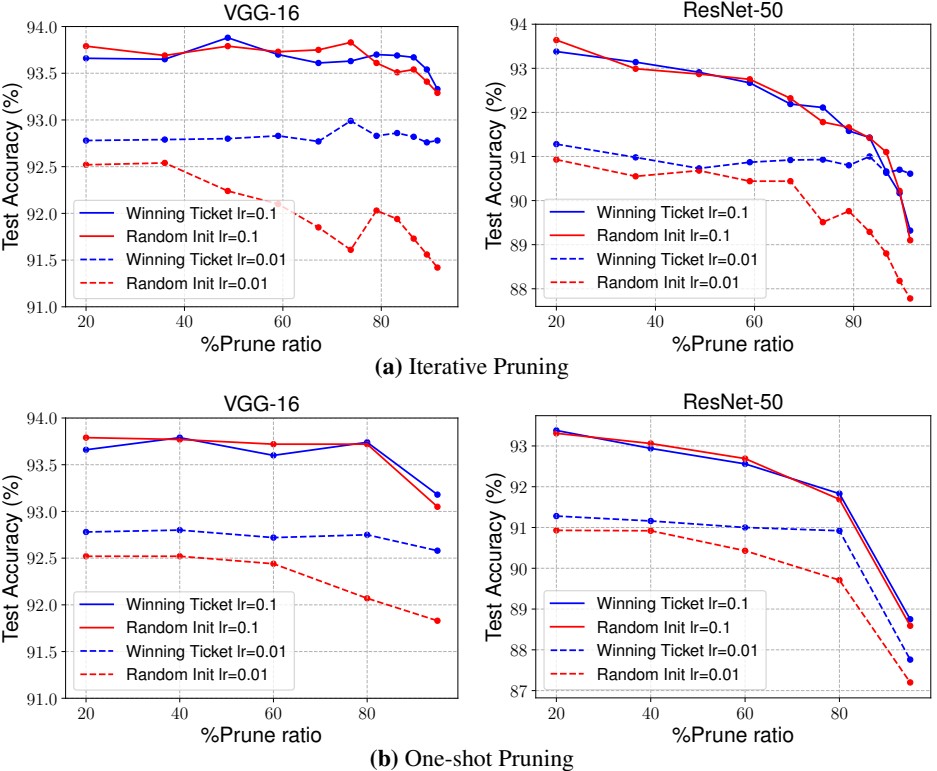

**(a)** Iterative Pruning

**(b)** One-shot Pruning

**Figure 7:** Comparisons with the Lottery Ticket Hypothesis (Frankle & Carbin, 2019) for iterative/one-shot unstructured pruning (Han et al., 2015) with two initial learning rates 0.1 and 0.01, on CIFAR-10 dataset. Each point is averaged over 5 runs. Using the winning ticket as initialization only brings improvement when the learning rate is small (0.01), however such small learning rate leads to a lower accuracy than the widely used large learning rate (0.1).

In this section, we show that the difference in learning rate is what causes the seemingly contradicting behaviors between our work and Frankle & Carbin (2019), in the case of unstructured pruning on CIFAR. For structured pruning, when using both large and small learning rates, the winning ticket does not outperform random initialization.

We test the Lottery Ticket Hypothesis by comparing the models trained with original initialization ("winning ticket") and that trained from randomly re-initialized weights. We experiment with two choices of initial learning rate (0.1 and 0.01) with a stepwise decay schedule, using momentum SGD. 0.1 is used in our previous experiments and most prior works on CIFAR and ImageNet. Following Frankle & Carbin (2019), we investigate both iterative pruning (prune 20% in each iteration) and one-shot pruning for unstructured pruning. We show our results for unstructured pruning (Han et al., 2015) in Figure 7 and Table 8, and $L_1$-norm based filter pruning (Li et al., 2017) in Table 9.

| Dataset | Model | Unpruned | Prune Ratio | Winning Ticket | Random Init |
|---|---|---|---|---|---|
| CIFAR-10 | VGG-16 | 93.76 ($\pm$0.20) | 20% | 93.66 ($\pm$0.20) | **93.79** ($\pm$0.11) |
| | | | 40% | **93.79** ($\pm$0.12) | 93.77 ($\pm$0.10) |
| | | | 60% | 93.60 ($\pm$0.13) | **93.72** ($\pm$0.11) |
| | | | 80% | **93.74** ($\pm$0.15) | 93.72 ($\pm$0.16) |
| | | | 95% | **93.18** ($\pm$0.12) | 93.05 ($\pm$0.21) |
| | ResNet-50 | 93.48 ($\pm$0.20) | 20% | **93.38** ($\pm$0.18) | 93.31 ($\pm$0.24) |
| | | | 40% | 92.94 ($\pm$0.12) | **93.06** ($\pm$0.22) |
| | | | 60% | 92.56 ($\pm$0.20) | **92.69** ($\pm$0.11) |
| | | | 80% | **91.83** ($\pm$0.20) | 91.69 ($\pm$0.21) |
| | | | 95% | **88.75** ($\pm$0.18) | 88.59 ($\pm$0.09) |

**(a)** One-shot pruning with initial learning rate 0.1

| Dataset | Model | Unpruned | Prune Ratio | Winning Ticket | Random Init |
|---|---|---|---|---|---|
| CIFAR-10 | VGG-16 | 92.69 ($\pm$0.12) | 20% | **92.78** ($\pm$0.11) | 92.52 ($\pm$0.15) |
| | | | 40% | **92.80** ($\pm$0.18) | 92.52 ($\pm$0.15) |
| | | | 60% | **92.72** ($\pm$0.16) | 92.44 ($\pm$0.19) |
| | | | 80% | **92.75** ($\pm$0.07) | 92.07 ($\pm$0.25) |
| | | | 95% | **92.58** ($\pm$0.25) | 91.83 ($\pm$0.11) |
| | ResNet-50 | 91.06 ($\pm$0.28) | 20% | **91.28** ($\pm$0.15) | 90.93 ($\pm$0.34) |
| | | | 40% | **91.16** ($\pm$0.07) | 90.92 ($\pm$0.10) |
| | | | 60% | **91.00** ($\pm$0.15) | 90.43 ($\pm$0.16) |
| | | | 80% | **90.92** ($\pm$0.08) | 89.71 ($\pm$0.18) |
| | | | 95% | **87.76** ($\pm$0.19) | 87.20 ($\pm$0.17) |

**(b)** One-shot pruning with initial learning rate 0.01

**Table 8:** Comparisons with the Lottery Ticket Hypothesis (Frankle & Carbin, 2019) for one-shot unstructured pruning (Han et al., 2015) with two initial learning rates: 0.1 and 0.01. The same results are visualized in Figure 7b. Using the winning ticket as initialization only brings improvement when the learning rate is small (0.01), however such small learning rate leads to a lower accuracy than the widely used large learning rate (0.1).

From Figure 7 and Table 8, we see that for unstructured pruning, using the original initialization as in (Frankle & Carbin, 2019) only provides advantage over random initialization with small initial learning rate 0.01. For structured pruning as Li et al. (2017), it can be seen from Table 9 that using the original initialization is only on par with random initialization for both large and small initial learning rates. In both cases, we can see that the small learning rate gives lower accuracy than the widely-used large learning rate. To summarize, in our evaluated settings, the winning ticket only brings improvement in the case of unstructured pruning, with small initial learning rate, but this small learning rate yields inferior accuracy compared with the widely-used large learning rate. Note that Frankle & Carbin (2019) also report in their Section 5, that the winning ticket cannot be found on ResNet-18/VGG using the large learning rate. The reason why the original initialization is helpful when the learning rate is small, might be the weights of the final trained model are not too far from the original initialization due to the small parameter updating stepsize.

| Dataset | Model | Unpruned | Pruned Model | Winning Ticket | Random Init |
|---------|-------|----------|--------------|----------------|-------------|
| CIFAR-10 | VGG-16 | 93.63 (±0.16) | VGG-16-A | **93.62** (±0.09) | 93.60 (±0.15) |
| | ResNet-56 | 93.14 (±0.12) | ResNet-56-A | 92.72 (±0.10) | **92.75** (±0.26) |
| | | | ResNet-56-B | 92.78 (±0.23) | **92.90** (±0.27) |
| | ResNet-110 | 93.14 (±0.24) | ResNet-110-A | **93.21** (±0.09) | **93.21** (±0.21) |
| | | | ResNet-110-B | 93.15 (±0.12) | **93.37** (±0.29) |

**(a)** Initial learning rate 0.1

| Dataset | Model | Unpruned | Pruned Model | Winning Ticket | Random Init |
|---------|-------|----------|--------------|----------------|-------------|
| CIFAR-10 | VGG-16 | 92.64 (±0.05) | VGG-16-A | 92.65 (±0.18) | **92.67** (±0.22) |
| | ResNet-56 | 89.81 (±0.27) | ResNet-56-A | **90.00** (±0.15) | 89.87 (±0.25) |
| | | | ResNet-56-B | 89.75 (±0.35) | **89.81** (±0.24) |
| | ResNet-110 | 89.43 (±0.39) | ResNet-110-A | 89.48 (±0.35) | **89.49** (±0.10) |
| | | | ResNet-110-B | **89.36** (±0.30) | 89.35 (±0.16) |

**(b)** Initial learning rate 0.01

**Table 9:** Experiments on the Lottery Ticket Hypothesis (Frankle & Carbin, 2019) on a structured pruning method ($L_1$-norm based filter pruning (Li et al., 2017)) with two initial learning rates: 0.1 and 0.01. In both cases, using winning tickets does not bring improvement on accuracy.

# B    RESULTS ON SOFT FILTER PRUNING (HE ET AL., 2018A)

**Soft Filter Pruning (SFP) (He et al., 2018a)** prunes filters every epoch during training but also keeps updating the pruned filters, i.e., the pruned weights have the chance to be recovered. In the original paper, SFP can either run upon a random initialized model or a pretrained model. It falls into the category of predefined methods (Figure 2). Table 10 shows our results without using pretrained models and Table 11 shows the results with a pretrained model. We use authors' code (He et al., 2018b) for obtaining the results. It can be seen that Scratch-E outperforms pruned models for most of the time and Scratch-B outperforms pruned models in nearly all cases. Therefore, our conclusion also holds on this method.

| Dataset | Model | Unpruned | Prune Ratio | Pruned | Scratch-E | Scratch-B |
|---------|-------|----------|-------------|--------|-----------|-----------|
| CIFAR-10 | ResNet-20 | 92.41 (±0.12) | 10% | 92.00 (±0.32) | **92.22** (±0.15) | 92.13 (±0.10) |
| | | | 20% | 91.50 (±0.30) | 91.62 (±0.12) | **91.67** (±0.15) |
| | | | 30% | 90.78 (±0.15) | 90.93 (±0.10) | **91.07** (±0.23) |
| | ResNet-32 | 93.22 (±0.16) | 10% | 93.28 (±0.05) | **93.42** (±0.40) | 93.08 (±0.13) |
| | | | 20% | 92.50 (±0.17) | 92.68 (±0.20) | **92.96** (±0.11) |
| | | | 30% | 92.02 (±0.11) | 92.37 (±0.12) | **92.56** (±0.06) |
| | ResNet-56 | 93.80 (±0.12) | 10% | 93.77 (±0.07) | 93.42 (±0.40) | **93.98** (±0.21) |
| | | | 20% | 93.14 (±0.42) | 93.44 (±0.05) | **93.71** (±0.14) |
| | | | 30% | 93.01 (±0.09) | 93.19 (±0.20) | **93.57** (±0.12) |
| | | | 40% | 92.59 (±0.14) | 92.80 (±0.25) | **93.07** (±0.25) |
| | ResNet-110 | 93.77 (±0.23) | 10% | 93.60 (±0.50) | **94.21** (±0.39) | 94.13 (±0.37) |
| | | | 20% | 93.63 (±0.44) | 93.52 (±0.18) | **94.29** (±0.18) |
| | | | 30% | 93.26 (±0.37) | 93.70 (±0.16) | **93.92** (±0.13) |
| ImageNet | ResNet-34 | 73.92 | 30% | 71.83 | 71.67 | **72.97** |
| | ResNet-50 | 76.15 | 30% | 74.61 | 74.98 | **75.56** |

**Table 10:** Results (accuracy) for Soft Filter Pruning (He et al., 2018a) without pretrained models.

| Dataset | Model | Unpruned | Prune Ratio | Pruned | Scratch-E | Scratch-B |
|---------|-------|----------|-------------|--------|-----------|-----------|
| CIFAR-10 | ResNet-56 | 93.80 (±0.12) | 30% | 93.51 (±0.26) | **94.45** (±0.30) | 93.77 (±0.25) |
| | | | 40% | 93.10 (±0.34) | **93.84** (±0.16) | 93.41 (±0.08) |
| | ResNet-110 | 93.77 (±0.23) | 30% | 93.46 (±0.19) | 93.89 (±0.17) | **94.37** (±0.24) |

**Table 11:** Results (accuracy) for Soft Filter Pruning (He et al., 2018a) using pretrained models.

## C    TRANSFER LEARNING TO OBJECT DETECTION

We have shown that for structured pruning the small pruned model can be trained from scratch to match the accuracy of the fine-tuned model in classification tasks. To see whether this phenomenon would also hold for transfer learning to other vision tasks, we evaluate the $L_1$-norm based filter pruning method (Li et al., 2017) on the PASCAL VOC object detection task, using Faster-RCNN (Ren et al., 2015).

Object detection frameworks usually require transferring model weights pre-trained on ImageNet classification, and one can perform pruning either before or after the weight transfer. More specifically, the former could be described as "train on classification, prune on classification, fine-tune on classification, transfer to detection", while the latter is "train on classification, transfer to detection, prune on detection, fine-tune on detection". We call these two approaches Prune-C (classification) and Prune-D (detection) respectively, and report the results in Table 12. With a slight abuse of notation, here Scratch-E/B denotes "train the small model on classification, transfer to detection", and is different from the setup of detection without ImageNet pre-training as in Shen et al. (2017).

| Dataset | Model | Unpruned | Pruned Model | Prune-C | Prune-D | Scratch-E | Scratch-B |
|---------|-------|----------|--------------|---------|---------|-----------|-----------|
| PASCAL VOC 07 | ResNet-34 | 71.69 | ResNet34-A | 71.47 | 70.99 | 71.64 | **71.90** |
| | | | ResNet34-B | 70.84 | 69.62 | **71.68** | 71.26 |

**Table 12:** Results (mAP) for pruning on detection task. The pruned models are chosen from Li et al. (2017). Prune-C refers to pruning on classifcation pre-trained weights, Prune-D refers to pruning after the weights are transferred to detection task. Scratch-E/B means pre-training the pruned model from scratch on classification and transfer to detection.

For this experiment, we adopt the code and default hyper-parameters from Yang et al. (2017), and use PASCAL VOC 07 trainval/test set as our training/test set. For backbone networks, we evaluate ResNet-34-A and ResNet-34-B from the $L_1$-norm based filter pruning (Li et al., 2017), which are pruned from ResNet-34. Table 12 shows our result, and we can see that the model trained from scratch can surpass the performance of fine-tuned models under the transfer setting.

Another interesting observation from Table 12 is that Prune-C is able to outperform Prune-D, which is surprising since if our goal task is detection, directly pruning away weights that are considered unimportant for detection should presumably be better than pruning on the pre-trained classification models. We hypothesize that this might be because pruning early in the classification stage makes the final model less prone to being trapped in a bad local minimum caused by inheriting weights from the large model. This is in line with our observation that Scratch-E/B, which trains the small models from scratch starting even earlier at the classification stage, can achieve further performance improvement.

## D    AGGRESSIVELY PRUNED MODELS

It would be interesting to see whether our observation still holds if the model is very aggressively pruned, since they might not have enough capacity to be trained from scratch and achieve decent accuracy. Here we provide results using Network Slimming (Liu et al., 2017) and $L_1$-norm based filter pruning (Li et al., 2017). From Table 13, Table 14 and Table 15, it can be seen that when the prune ratio is large, training from scratch is better than fine-tuned models by an even larger margin, and in this case fine-tuned models are significantly worse than the unpruned models. Note that in Table 2, the VGG-Tiny model we evaluated for ThiNet (Luo et al., 2017) is also a very aggressively pruned model (FLOPs reduced by $15\times$ and parameters reduced by $100\times$).

| Dataset | Model | Unpruned | Prune Ratio | Fine-tuned | Scratch-E | Scratch-B |
|---------|-------|----------|-------------|------------|-----------|-----------|
| CIFAR-10 | PreResNet-56 | 93.69 (±0.07) | 80% | 74.66 (±0.96) | 88.25 (±0.38) | **88.65** (±0.32) |
| | PreResNet-164 | 95.04 (±0.16) | 80% | 91.76 (±0.38) | 93.21 (±0.17) | **93.49** (±0.20) |
| | | | 90% | 82.06 (±0.92) | 87.55 (±0.68) | **88.44** (±0.19) |
| | DenseNet-40 | 94.10 (±0.12) | 80% | 92.64 (±0.12) | 93.07 (±0.08) | **93.61** (±0.12) |
| CIFAR-100 | DenseNet-40 | 73.82 (±0.34) | 80% | 69.60 (±0.22) | 71.04 (±0.36) | **71.45** (±0.30) |

**Table 13:** Results (accuracy) for Network Slimming (Liu et al., 2017) when the models are aggressively pruned. "Prune ratio" stands for total percentage of channels that are pruned in the whole network. Larger ratios are used than the original paper of Liu et al. (2017).

| Dataset | Model | Unpruned | Prune Ratio | Fine-tuned | Scratch-E | Scratch-B |
|---------|-------|----------|-------------|------------|-----------|-----------|
| CIFAR-10 | ResNet-56 | 93.14 (±0.12) | 90% | 89.17 (±0.08) | 91.02 (±0.12) | **91.93** (±0.26) |

**Table 14:** Results (accuracy) for $L_1$-norm based filter pruning (Li et al., 2017) when the models are aggressively pruned.

| Dataset | Model | Unpruned | Prune Ratio | Fine-tuned | Scratch-E | Scratch-B |
|---------|-------|----------|-------------|------------|-----------|-----------|
| CIFAR-10 | VGG-19 | 93.50 (±0.11) | 95% | 93.34 (±0.13) | 93.21 (±0.17) | **93.63** (±0.18) |
| CIFAR-100 | VGG-19 | 71.70 (±0.31) | 95% | 70.22 (±0.38) | 70.88 (±0.35) | **72.08** (±0.15) |

**Table 15:** Results (accuracy) for unstructured pruning (Han et al., 2015) when the models are aggressively pruned.

# E    EXTENDING FINE-TUNING EPOCHS

Generally, pruning algorithms use fewer epochs for fine-tuning than training the large model (Li et al., 2017; He et al., 2017b; Luo et al., 2017). For example, $L_1$-norm based filter pruning (Li et al., 2017) uses 164 epochs for training on CIFAR-10 datasets, and only fine-tunes the pruned networks for 40 epochs. This is due to that mostly small learning rates are used for fine-tuning to better preserve the weights from the large model. Here we experiment with fine-tuning for more epochs (e.g., for the same number of epochs as Scratch-E) and show it does not bring noticeable performance improvement.

| Dataset | Model | Pruned Model | Fine-tune-40 | Fine-tune-80 | Fine-tune-160 | Scratch-E |
|---------|-------|--------------|--------------|--------------|---------------|-----------|
| CIFAR-10 | VGG-16 | VGG-16-A | 93.40 (±0.12) | 93.45 (±0.06) | 93.45 (±0.08) | **93.62** (±0.11) |
| | ResNet-56 | ResNet-56-A | **92.97** (±0.17) | 92.92 (±0.15) | 92.94 (±0.16) | 92.96 (±0.26) |
| | | ResNet-56-B | 92.68 (±0.19) | 92.67 (±0.14) | **92.76** (±0.16) | 92.54 (±0.19) |
| | ResNet-110 | ResNet-110-A | 93.14 (±0.16) | 93.12 (±0.19) | 93.04 (±0.22) | **93.25** (±0.29) |
| | | ResNet-110-B | 92.69 (±0.09) | 92.75 (±0.15) | 92.76 (±0.16) | **92.89** (±0.43) |

**Table 16:** "Fine-tune-40" stands for fine-tuning 40 epochs and so on. Scratch-E models are trained for 160 epochs. We observe that fine-tuning for more epochs does not help improve the accuracy much, and models trained from scratch can still perform on par with fine-tuned models.

We use $L_1$-norm filter pruning (Li et al., 2017) for this experiment. Table 16 shows our results with different number of epochs for fine-tuning. It can be seen that fine-tuning for more epochs gives negligible accuracy increase and sometimes small decrease, and Scratch-E models are still on par with models fine-tuned for enough epochs.

# F    EXTENDING THE STANDARD TRAINING SCHEDULE

In our experiments, we use the standard training schedule for both CIFAR (160 epochs) and ImageNet (90 epochs). Here we show that our observation still holds after we extend the standard training schedule. We use $L_1$-norm based filter pruning (Li et al., 2017) for this experiment. Table 17 shows our results when we extend the standard training schedule of CIFAR from 160 to 300 epochs. We observe that scratch trained models still perform at least on par with fine-tuned models.

| Dataset | Model | Unpruned | Pruned Model | Fine-tuned | Scratch-E | Scratch-B |
|---------|-------|----------|--------------|------------|-----------|-----------|
| CIFAR-10 | VGG-16 | 93.79 (±0.05) | VGG-16-A | 93.67 (±0.11) | 93.74 (±0.14) | **93.80** (±0.09) |
| | ResNet-56 | 93.52 (±0.05) | ResNet-56-A | 93.44 (±0.15) | 93.34 (±0.17) | **93.56** (±0.09) |
| | | | ResNet-56-B | 93.12 (±0.20) | 93.14 (±0.21) | **93.30** (±0.17) |
| | ResNet-110 | 93.82 (±0.32) | ResNet-110-A | 93.75 (±0.24) | 93.80 (±0.15) | **94.10** (±0.12) |
| | | | ResNet-110-B | 93.36 (±0.28) | 93.75 (±0.16) | **93.90** (±0.17) |

**Table 17:** Results for $L_1$-norm filter pruning (Li et al., 2017) when the training schedule of the large model is extended from 160 to 300 epochs.

## G    WEIGHT DISTRIBUTIONS

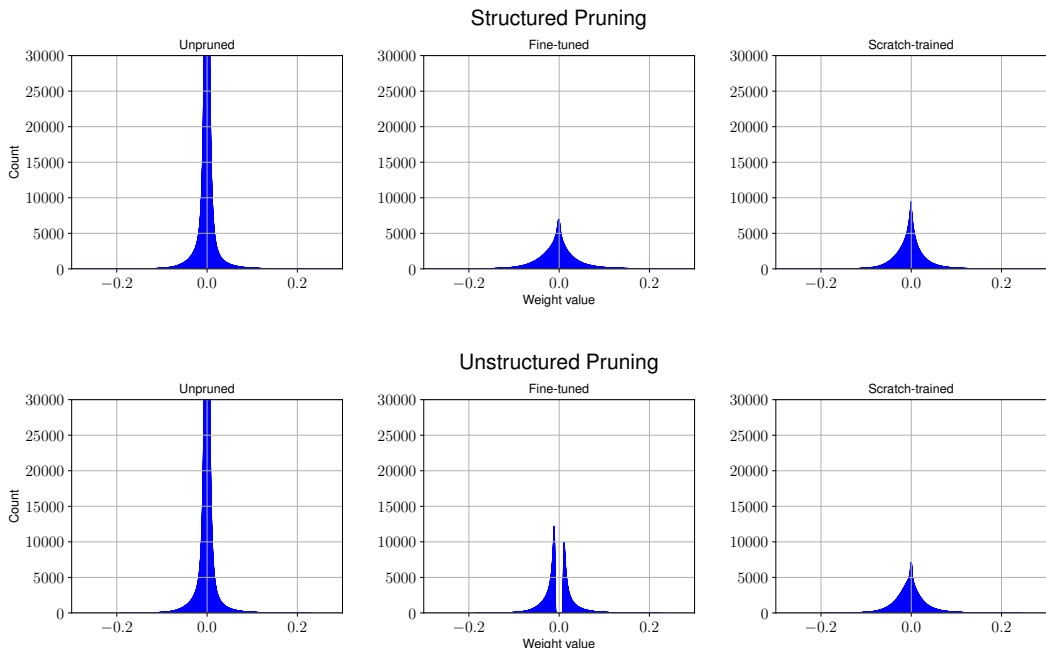

**Figure 8:** Weight distribution of convolutional layers for different pruning methods. We use VGG-16 and CIFAR-10 for this visualization. We compare the weight distribution of unpruned models, fine-tuned models and scratch-trained models. *Top*: Results for Network Slimming (Liu et al., 2017). *Bottom*: Results for unstructured pruning (Han et al., 2015).

Accompanying the discussion in subsection 4.3, we show the weight distribution of unpruned large models, fine-tuned pruned models and scratch-trained pruned models, for two pruning methods: (structured) Network Slimming (Liu et al., 2017) and unstructured pruning (Han et al., 2015). We choose VGG-16 and CIFAR-10 for visualization and compare the weight distribution of unpruned models, fine-tuned models and scratch-trained models. For Network Slimming, the prune ratio is 50%. For unstructured pruning, the prune ratio is 80%. Figure 8 shows our result. We can see that the weight distribution of fine-tuned models and scratch-trained pruned models are different from the unpruned large models – the weights that are close to zero are much fewer. This seems to imply that there are less redundant structures in the found pruned architecture, and support the view of architecture learning for automatic pruning methods.

For unstructured pruning, the fine-tuned model also has significantly different weight distribution from the scratch-trained model – it has nearly no close-to-zero values. This might be a potential reason why sometimes models trained from scratch cannot achieve the accuracy of the fine-tuned models, as shown in Table 6.

## H  MORE SPARSITY PATTERNS FOR PRUNED ARCHITECTURES

In this section we provide the additional results on sparsity patterns for the pruned models, accompanying the discussions of "More Analysis" in Section 5.

|         | 10%   | 20%   | 30%   | 40%   | 50%   | 60%   | 70%   |
|---------|-------|-------|-------|-------|-------|-------|-------|
| Stage 1 | 0.879 | 0.729 | 0.557 | 0.484 | 0.421 | 0.349 | 0.271 |
| Stage 2 | 0.959 | 0.863 | 0.754 | 0.651 | 0.537 | 0.428 | 0.320 |
| Stage 3 | 0.889 | 0.798 | 0.716 | 0.610 | 0.507 | 0.403 | 0.301 |

**Table 18:** Sparsity patterns of PreResNet-164 pruned on CIFAR-10 by Network Slimming shown in Figure 5 (left) under different prune ratio. The top row denotes the total prune ratio. The values denote the ratio of channels to be kept. We can observe that for a certain prune ratio, the sparsity patterns are close to uniform (across stages).

|         | 10% | | | 50% | | | 90% | | |
|---------|-------|-------|-------|-------|-------|-------|-------|-------|-------|
|         | 0.905 | 0.905 | 0.909 | 0.530 | 0.561 | 0.538 | 0.129 | 0.171 | 0.133 |
| Stage 1 | 0.900 | 0.912 | 0.899 | 0.559 | 0.588 | 0.551 | 0.166 | 0.217 | 0.176 |
|         | 0.903 | 0.913 | 0.902 | 0.532 | 0.563 | 0.547 | 0.142 | 0.172 | 0.163 |
|         | 0.906 | 0.911 | 0.906 | 0.485 | 0.523 | 0.503 | 0.073 | 0.102 | 0.085 |
| Stage 2 | 0.912 | 0.911 | 0.915 | 0.508 | 0.529 | 0.525 | 0.099 | 0.114 | 0.111 |
|         | 0.911 | 0.916 | 0.912 | 0.502 | 0.529 | 0.519 | 0.080 | 0.113 | 0.096 |
|         | 0.901 | 0.904 | 0.900 | 0.454 | 0.475 | 0.454 | 0.043 | 0.059 | 0.048 |
| Stage 3 | 0.885 | 0.891 | 0.889 | 0.409 | 0.420 | 0.415 | 0.032 | 0.033 | 0.035 |
|         | 0.898 | 0.903 | 0.902 | 0.450 | 0.468 | 0.458 | 0.042 | 0.055 | 0.046 |

**Table 19:** Average sparsity patterns of 3×3 kernels of PreResNet-110 pruned on CIFAR-100 by unstructured pruning shown in Figure 5 (middle) under different prune ratio. The top row denotes the total prune ratio. The values denote the ratio of weights to be kept. We can observe that for a certain prune ratio, the sparsity patterns are close to uniform (across stages).

|         | 10% | | | 50% | | | 90% | | |
|---------|-------|-------|-------|-------|-------|-------|-------|-------|-------|
|         | 0.861 | 0.856 | 0.858 | 0.507 | 0.495 | 0.510 | 0.145 | 0.129 | 0.142 |
| Stage 1 | 0.843 | 0.844 | 0.851 | 0.484 | 0.486 | 0.479 | 0.123 | 0.115 | 0.126 |
|         | 0.850 | 0.854 | 0.857 | 0.509 | 0.490 | 0.511 | 0.136 | 0.131 | 0.147 |
|         | 0.907 | 0.905 | 0.906 | 0.498 | 0.487 | 0.499 | 0.099 | 0.088 | 0.100 |
| Stage 2 | 0.892 | 0.888 | 0.892 | 0.442 | 0.427 | 0.444 | 0.064 | 0.043 | 0.065 |
|         | 0.907 | 0.906 | 0.905 | 0.497 | 0.485 | 0.493 | 0.095 | 0.082 | 0.098 |
|         | 0.897 | 0.901 | 0.899 | 0.470 | 0.475 | 0.472 | 0.060 | 0.060 | 0.064 |
| Stage 3 | 0.888 | 0.890 | 0.889 | 0.433 | 0.437 | 0.435 | 0.040 | 0.040 | 0.042 |
|         | 0.898 | 0.900 | 0.899 | 0.473 | 0.477 | 0.473 | 0.060 | 0.061 | 0.063 |

**Table 20:** Average sparsity patterns of 3×3 kernels of DenseNet-40 pruned on CIFAR-100 by unstructured pruning shown in Figure 5 (right) under different prune ratio. The top row denotes the total prune ratio. The values denote the ratio of weights to be kept. We can observe that for a certain prune ratio, the sparsity patterns are close to uniform (across stages).

|         | 10%   | 20%   | 30%   | 40%   | 50%   | 60%   |
|---------|-------|-------|-------|-------|-------|-------|
| Stage 1 | 0.969 | 0.914 | 0.883 | 0.875 | 0.844 | 0.836 |
| Stage 2 | 1.000 | 1.000 | 1.000 | 1.000 | 1.000 | 1.000 |
| Stage 3 | 0.991 | 0.975 | 0.966 | 0.957 | 0.947 | 0.947 |
| Stage 4 | 0.861 | 0.718 | 0.575 | 0.446 | 0.312 | 0.258 |
| Stage 5 | 0.871 | 0.751 | 0.626 | 0.486 | 0.352 | 0.132 |

**Table 21:** Sparsity patterns of VGG-16 pruned on CIFAR-10 by Network Slimming shown in Figure 3 (left) under different prune ratio. The top row denotes the total prune ratio. The values denote the ratio of channels to be kept. For each prune ratio, the latter stages tend to have more redundancy than earlier stages.

