# OpenReview forum: "Rethinking the Value of Network Pruning"
_ICLR.cc/2019/Conference_

### Official Review · AnonReviewer1 · 2018-11-01
**interesting paper, more in-depth analysis to support their findings would be better.**

**Rating:** 7
**Confidence:** 5

**Review:**

This paper reinvestigate several recent works on network pruning and find that the common belief about the necessity to train a large network before pruning may not hold. The authors find that training the pruned model from scratch can achieve similar, if not better, performance given enough time of training. Based on these observations, the author conclude that training a larger model followed by pruning is not necessary for obtaining an efficient model with similar performance. In other words, the pruned architecture is more important than the weights inherited from the large model. It reminds researchers to perform stronger baselines before showing complex pruning methods.

The paper is well organized and written. It re-evaluate the recent progresses made on this topic. Instead of comparing approaches by simply using the numbers from previous paper, the authors perform extensive experiments to verify whether training the pruned network from scratch would work. The results are very interesting, it suggests the researchers to tune the baseline “hardly” and stick to simple approach. However, here are some places that I have concerns with:

1. The two “common beliefs” actually state one thing, that is the weights of a pre-trained larger model can potentially help optimization for a smaller model.

2. I don’t quite agree with that “training” is the first step of a pruning pipeline as illustrated in Figure 1.  Actually the motivation or the common assumption for pruning is that there are already existing trained models (training is already finished) with good performance. If a trained model does not even exist, then one can certainly train various thin/smaller model from scratch as before, this is still a trial and error process.

3. “The value of pruning”. The goal of pruning is to explore a “thin” or “shallower” version of it with similar accuracy while avoiding the exhaustive architecture search with heavy training processes. Thus the first value of pruning is to explore efficient architecture while avoiding heavy training. Therefore, it should be fast and efficient, ideally with no retraining or little fine-tuning. When the pruning method is too complex to implement or requires much more time than training from scratch, it could be an overkill and adds little value, especially when the performance is not better enough. Therefore, it is more informative if the authors would report the time/complexities for pruning/fine-tuning .

4. The second value of pruning lies at understand the redundancy of the model and providing insights for more efficient architecture designs.

5. Comparing to random initialization, pruning simply provide an initialization point inherited from the larger network. The essential question the author asked is whether a subset of pre-trained weights can outperform random initialization. This seems to be a common belief in transfer learning, knowledge distillation and the studies on initialization. The authors conclude that the accuracy of an architecture is determined by the architecture itself, but not the initialization. If this is true, training from scratch should have similar (but not better) result as fine-tuning a pruned model.  As the inherited weights can also be viewed as a “random” initialization. Both methods should reach equivalent good solution if they are trained with enough number of epochs. Can this be verified with experiments?

6. The experiments might not be enough to reject the common belief. The experiments only spoke that the pruned architectures can still be easily trained and encounter no difficulties during the optimization. One conjecture is that the pruned models in the previous work still have enough capacity for keeping good accuracy. What if the models are significantly pruned (say more than 70% of channels got pruned), is training from scratch still working well? It would add much value if the author can identify when training from scratch fails to match the performance obtained by pruning and fine-tuning.

7. In Section 4.1, “scratch-trained models achieve at least the same level of accuracy as fine-tuned models”. First, the ResNet-34-pruned A/B for this comparison does not have significant FLOPs reduction (10% and 24% FLOPs reduction). Fine-tuning still has advantage as it only takes ¼ of training time compare to scratch-E. Second, it is interesting that fine-tuning has generally smaller variance than stratch-E (except VGG-19). Would this imply that fine-tuning a pruned model produce more stable result? It would be more complete if there is variance analysis for the imagenet result.

8. What is the training/fine-tuning hyperparameters used in section 4.1?  Note that in the experiment of Li et al, 2017, scratch-E takes 164 epochs to train from scratch, while fine-tuning takes only 40 epochs. Like suggested above, if we fine-tune it with more epochs, would it achieve equivalent performance? Also, what is the hyperparameter used in scratch-E? Note that the original paper use batch size 128. If the authors adopts a smaller batch-size for scratch-E, then it has in more iterations and could certainly result in better performance according to recent belief that small batch-size generates better.

9. The conclusion of section 5 is not quite clear or novel. Using uniform pruning ratio for pruning is expected to perform worse than automatic pruning methods as it does not consider the importance difference of each layer and. This comes back to my point 3 & 4 about the value of pruning, that is the value of pruning lies at the analysis of the redundancy of the network. There are a number of works worked on analyzing the importance of different layers of filters. So I think the “hypothesis” of “the value of automatic pruning methods actually lies in the resulting architecture rather than the inherited weight” is kind of straightforward. Also, why not use FLOPs as x-axis in Figure 3?


Minor: It might be more accurate to use “L1-norm based Filter Pruning (Li et al., 2017)” as literally “channels” usually refers to feature maps, which are by-products of the model but not the model itself.

I  will revise my score if authors can address above concerns.


--------- review after rebuttal----------
#1#2 It would be great if the authors can make it clear that training is not the always the first step and the value of pruning in introduction rather than mentioning in conclusion. Saving training time is still an important factor when training from scratch is expensive.

#5 “fine-tuning with enough epochs”.
I understand that the authors are mainly questioning about whether training from scratch is necessarily bad than pruning and fine-tuning. The author do find that “training from scratch is better when the number of epochs is large enough”. But we see that fine-tuning ResNet-56 A/B with 20 epochs does outperform (or is equivalent to) scratch training for the first 160 epochs, which validates “fine-tuning is faster to converge”.  However, training 320 epochs (16x more comparing to 20 epochs fine-tuning and 2x comparing with normal training from scratch) is not quite coherent with the setting of “scratch B”, as ResNet-56 B just reduce 27% FLOPs.

The other part of the question is still unclear, i.e., the author claimed that the accuracy of an architecture is determined by the architecture itself, but not the initialization, then both fine-tuning and scratch training should reach equivalent solution if they are well trained enough, regardless of the initialization or pruning method. The learning rate for scratch training is already well known (learning rate drop brings boost the accuracy). However, learning rate schedule for fine-tuning (especially for significantly pruned model as for reply#6) is not well explored. I wonder whether that a carefully tuned learning rate/hyperparameters for fine-tuning may get the same or better performance as scratch training.

Questions:
- Are both methods using the same learning rate schedule between epoch 160 and epoch 320?
- The ResNets-56 A/B results in the reply#8 does not match the reported performance in reply#5. e.g., it shows 92.67(0.09) for ResNet-56-B with 40-epochs fine-tuning in reply5,  but it turns out to be 92.68(±0.19) in reply#8.
- It would be great if the authors can add convergence curves for fine-tuning and scratch training for easier comparison.


#6 The failure case for sparse pruning on ImageNet is interesting and it would be great to have the imageNet result reported and discussed.

The authors find that “when the pruned ratio is large enough, training from scratch is better by a even larger margin than fine-tuning”.  This could be due to following reasons:
      1. When the pruning ratio is large, the pruned model with preserved weights is significantly different from the original model, and fine-tuning with small learning rate and limited number of epochs is not enough to recover the accuracy. As mentioned earlier, tuning the hyperparameters for fine-tuning based on pruning ratio might improve the performance of fine-tuning.
      2. Though the pruning ratio is large, the model used in this experiment may still have large capacity to reach good performance. How about pruning ResNet-56 with significant pruning ratios?

Finally, based on above observations, it seems to me that the preserved weights is more essential for fast fine-tuning but less useful for significant pruning ratios.

-------- update ----------------

The authors addressed most of my concerns. Some questions are still remaining in my comment “Review after rebuttal”,  specifically, fine-tuning a pruned network may still get good performance if the hyperparameters are carefully tuned based on the pruning ratios, or in other words, the preserved weights is more essential for fast fine-tuning but less useful for significant pruning ratios. The authors may need to carefully made the conclusion from the observations. I would hope the authors can address these concerns in the future version.

However, I think the paper is overall well-written and existing content is inspiring enough for readers to further explore the trainability of the pruned network. Therefore I raised my score to 7.

---

> ### Author Response · Authors · 2018-11-14
> **Response to AnonReviewer1 [4/4]**
>
> For fine-tuning epochs, we have run experiments to show that more epochs don’t improve fine-tuning noticeably. This is because very small learning rate is used to preserve the inherited weights, as also mentioned in point 5. The results are as follows:
> ------------------------------------------------------------------------------------------
> Pruned Model     Fine-tune-40        Fine-tune-80       Fine-tune-160
> ------------------------------------------------------------------------------------------
> VGG-16               93.40(±0.12)        93.45(±0.06)         93.45(±0.08)
> ResNet-56-A       92.97(±0.17)        92.92(±0.15)         92.94(±0.16)
> ResNet-56-B       92.68(±0.19)        92.67(±0.14)         92.76(±0.16)
> ResNet-110-A     93.14(±0.16)        93.12(±0.19)         93.04(±0.22)
> ResNet-110-B     92.69(±0.09)        92.75(±0.15)         92.76(±0.16)
> ------------------------------------------------------------------------------------------
> It can be seen that fine-tuning for more epochs gives negligible accuracy increase and sometimes small decrease.
>
> 9. ## Conclusion of Section 5 ## Yes, we agree with your points in 3&4, and our experiments in Section 4 and 5 are to verify these points. We also agree that the conclusion of Section 5 may seem straightforward to some audience, but we think pruning is not very widely recognized as architecture search. Conventional network pruning and architecture search works still use totally different techniques, with the former focus on selecting important weights from a larger network and the later typically uses reinforcement learning or evolutionary algorithms to search an architecture through iterations. Pruning is usually mentioned as a model compression technique, in a resource-saving context, instead of being treated as an architecture search method.
>
> To our knowledge, our work is one of the first to draw a distinction between predefined and automatic pruning methods, and also one of the first to compare automatically pruned architecture with uniform pruning. Previous works compare pruned and fine-tuned model with the original large model, this is not sufficient to for "pruning can be seen as architecture search": 1. The benefit could be from the inherited weights, not the architecture. 2. Comparison with uniform pruning is missing. In Section 4, we show that the performance of training from scratch can be on par with pruning, however, comparison with uniform pruning is still needed. In Section 5, we break the tie between inherited weights and the resulting architecture, training the pruned architecture from scratch and comparing with uniform pruning. This provides further evidence that the value lies in searching efficient architecture.
>
> Also, in Section 5, we have shown that we can transfer the sparsity pattern in the pruned model to a different architecture on a different dataset. This implies that in these cases, we don’t need to train a large model on the target dataset to find the efficient model and transferred design patterns can help us design an efficient model from scratch. This experiment was also not investigated in prior works and the conclusion is not obvious. We will include more results on this point in the revision.
>
> We will also add some discussions about the differences and similarities between pruning as architecture and conventional architecture search. We will mention that some previous works have made connections between pruning and architecture search as well.
>
> The main reason why we didn’t include figures with FLOPs as x-axis is mainly the space limit. We have included figures with FLOPs as x-axis in Section 2 of this anonymous pdf link ( https://drive.google.com/open?id=1BjGJQASV-CuGoq-nVErIRihHMdVwCZxl ).
>
> Minor: Thank you for reminding, we will change the name to "filter pruning" as suggested.
>
> We will upload a revision after we address other reviewers' concerns, and please advise on which part of this response you would like to see to be reflected in the revision (other than the content we already plan to include). Again, thank you for your detailed review. If you have any further questions, we are happy to answer.
>
>
> [1]  Channel Pruning for Accelerating Very Deep Neural Networks. He et al., ICCV 2017.
> [2]  ThiNet: A Filter Level Pruning Method for Deep Neural Network Compression. Luo et al., ICCV 2017.
> [3]  Learning Efficient Convolutional Networks through Network Slimming. Liu et al., ICCV 2017.
> [4]  Pruning Filters for Efficient ConvNets. Li et al., ICLR 2017.
> [5]  Learning both Weights and Connections for Efficient Neural Networks. Han et al., NIPS 2015.
> [6]  Deep Compression: Compressing Deep Neural Networks with Pruning, Trained Quantization and Huffman Coding. Han et al., ICLR 2016.
> [7]  Data-Driven Sparse Structure Selection for Deep Neural Networks. Huang et al., ECCV 2018.
> [8]  Deep Residual Learning for Image Recognition. He et al., CVPR 2016.
> [9]  Densely Connected Convolutional Networks. Liu et al., CVPR 2017.

---

> > ### Comment · AnonReviewer1 · 2018-11-26
> > **Review after rebuttal**
> >
> > Thanks for the detailed reply and additional experiments. Just had more comments and questions
> >
> > #1#2 It would be great if the authors can make it clear that training is not the always the first step and the value of pruning in introduction rather than mentioning in conclusion. Saving training time is still an important factor when training from scratch is expensive.
> >
> > #5 “fine-tuning with enough epochs”.
> > I understand that the authors are mainly questioning about whether training from scratch is necessarily bad than pruning and fine-tuning. The author do find that “training from scratch is better when the number of epochs is large enough”. But we see that fine-tuning ResNet-56 A/B with 20 epochs does outperform (or is equivalent to) scratch training for the first 160 epochs, which validates “fine-tuning is faster to converge”.  However, training 320 epochs (16x more comparing to 20 epochs fine-tuning and 2x comparing with normal training from scratch) is not quite coherent with the setting of “scratch B”, as ResNet-56 B just reduce 27% FLOPs.
> >
> > The other part of the question is still unclear, i.e., the author claimed that the accuracy of an architecture is determined by the architecture itself, but not the initialization, then both fine-tuning and scratch training should reach equivalent solution if they are well trained enough, regardless of the initialization or pruning method. The learning rate for scratch training is already well known (learning rate drop brings boost the accuracy). However, learning rate schedule for fine-tuning (especially for significantly pruned model as for reply#6) is not well explored. I wonder whether that a carefully tuned learning rate/hyperparameters for fine-tuning may get the same or better performance as scratch training.
> >
> > Questions:
> > - Are both methods using the same learning rate schedule between epoch 160 and epoch 320?
> > - The ResNets-56 A/B results in the reply#8 does not match the reported performance in reply#5. e.g., it shows 92.67(0.09) for ResNet-56-B with 40-epochs fine-tuning in reply5,  but it turns out to be 92.68(±0.19) in reply#8.
> > - It would be great if the authors can add convergence curves for fine-tuning and scratch training for easier comparison.
> >
> >
> > #6 The failure case for sparse pruning on ImageNet is interesting and it would be great to have the imageNet result reported and discussed.
> >
> > The authors find that “when the pruned ratio is large enough, training from scratch is better by a even larger margin than fine-tuning”.  This could be due to following reasons:
> >       1. When the pruning ratio is large, the pruned model with preserved weights is significantly different from the original model, and fine-tuning with small learning rate and limited number of epochs is not enough to recover the accuracy. As mentioned earlier, tuning the hyperparameters for fine-tuning based on pruning ratio might improve the performance of fine-tuning.
> >       2. Though the pruning ratio is large, the model used in this experiment may still have large capacity to reach good performance. How about pruning ResNet-56 with significant pruning ratios?
> >
> > Finally, based on above observations, it seems to me that the preserved weights is more essential for fast fine-tuning but less useful for significant pruning ratios.

---

> > > ### Author Response · Authors · 2018-12-04
> > > **Response to AnonReviewer1 [3/3]**
> > >
> > >
> > > 2) #ResNet-56 Results# We have added results for large prune ratios for Network Slimming on PreResNet-56 and L1-norm filter pruning on ResNet-56. For PreResNet-56, the prune ratio is 80%. For ResNet-56, we use uniform pruning ratio 90% for each layer. As before, we present results for both the original fine-tuning schedule (denoted as ‘Fine-tune’) and scratch-training/fine-tuning with restart (denoted as ‘Fine-tune-restart’).
> > >
> > > Network Slimming:
> > > ------------------------------------------------------------------------------------------------------------------------------
> > >   Dataset          Model      Ratio    Fine-tune      Scratch-E       Scratch-B    Fine-tune-restart
> > > ------------------------------------------------------------------------------------------------------------------------------
> > > CIFAR-10  PreResNet-56  80%  74.66(±0.96)  88.25(±0.38)   88.65(±0.32)       86.71(±1.23)
> > > ------------------------------------------------------------------------------------------------------------------------------
> > >
> > > ------------------------------------------------------------------------------------------------------------------------------
> > >   Dataset          Model        Ratio     Fine-tune-restart    Scratch-E-restart     Scratch-B-restart
> > > ------------------------------------------------------------------------------------------------------------------------------
> > > CIFAR-10  PreResNet-56     80%        86.71(±1.23)           88.61(±0.62)             88.64(±0.28)
> > > ------------------------------------------------------------------------------------------------------------------------------
> > >
> > > L1-norm filter pruning:
> > > ------------------------------------------------------------------------------------------------------------------------------
> > >  Dataset          Model      Ratio      Fine-tune      Scratch-E        Scratch-B      Fine-tune-restart
> > > ------------------------------------------------------------------------------------------------------------------------------
> > > CIFAR-10      ResNet-56    90%      89.17(0.08)   91.02(±0.12)   91.93(±0.26)      90.29(±0.26)
> > > ------------------------------------------------------------------------------------------------------------------------------
> > >
> > > ------------------------------------------------------------------------------------------------------------------------------
> > >   Dataset          Model        Ratio    Fine-tune-restart    Scratch-E-restart     Scratch-B-restart
> > > ------------------------------------------------------------------------------------------------------------------------------
> > > CIFAR-10      ResNet-56       90%       90.29(±0.26)            91.57(±0.10)            91.40(±0.34)
> > > ------------------------------------------------------------------------------------------------------------------------------
> > > It can be seen that for both pruning methods, training significantly pruned models (even without learning rate restart) from scratch can still outperform fine-tuned models. This supports that "the preserved weights is more essential for fast fine-tuning but less useful for significant pruning ratios".
> > >
> > > Thanks again for your questions and any further discussions and suggestions are welcome.
> > >
> > > [1] SGDR: Stochastic Gradient Descent with Warm Restarts. Loshchilov et al., ICLR 2017.

---

> > > ### Author Response · Authors · 2018-12-04
> > > **Response to AnonReviewer1 [2/3]**
> > >
> > >
> > > (Continuing) Since in the table above fine-tuning takes 160 epochs which is much more than the original 40 epochs, now we consider the epochs for both large model training and fine-tuning when determining scratch-training epochs (our focus is when there are no pretrained large models). We also use the same learning rate restart schedule and compare this with "finetune-restart" in the table below (the learning rate restart is shown to be beneficial for performance in [1]). It can be seen that Scratch-E/B with restart can still perform comparably with this better fine-tuning learning rate schedule.
> > >
> > > ------------------------------------------------------------------------------------------------------------------------
> > >   Dataset      Pruned  Model  Fine-tune-restart   Scratch-E-restart    Scratch-B-restart
> > > ------------------------------------------------------------------------------------------------------------------------
> > > CIFAR-10       VGG-16-A              93.80(±0.07)           93.75(±0.21)             93.84(±0.18)
> > > CIFAR-10      ResNet-56-A          93.46(±0.21)           93.44(±0.08)             93.27(±0.12)
> > > CIFAR-10      ResNet-56-B          93.29(±0.19)           93.11(±0.10)             93.36(±0.29)
> > > CIFAR-10     ResNet-110-A         93.55(±0.17)           93.84(±0.11)             93.56(±0.32)
> > > CIFAR-10     ResNet-110-B         93.51(±0.15)           93.54(±0.26)             93.58(±0.51)
> > > ------------------------------------------------------------------------------------------------------------------------
> > >
> > > This set of experiments demonstrates that a better choice of learning rate schedule for fine-tuning can boost the accuracy (possibly due to more epochs and the lr "restart" effect [1]), but we still can train the pruned model from scratch and achieve comparable performance. This is in line with our conclusion that training a large model is not absolutely necessary.
> > >
> > > 3) (Q2) This slight result difference is because the two sets of experiments in reply#8 and reply#5 are done independently.
> > >
> > > 4) (Q3) We have plotted some convergence curves (including significantly pruned models, with or without learning rate restart), in Section 2 of this pdf link https://drive.google.com/open?id=1dnMDj_kAYblUjHPm9CAGi3bztkyH5dsj . We will include some of the convergence curves in the next revision.
> > >
> > > #6-1 ## Failure case on ImageNet ## Thanks for the suggestion. In the newest revision, we have added this result and some discussions in Section 4.2 (Table 6) and Appendix G.
> > >
> > > #6-2 ## Significantly pruned models##
> > > 1) #Fine-tuning learning rate# We present the results for significantly pruned models (the models we evaluated in our last response) when considering the better restart learning rate schedule for fine-tuning:
> > > -------------------------------------------------------------------------------------------------------------------------------
> > >   Dataset           Model        Ratio      Fine-tune       Scratch-E       Scratch-B    Fine-tune-restart
> > > -------------------------------------------------------------------------------------------------------------------------------
> > > CIFAR-10     DenseNet-40     80%     92.64(±0.12)  93.07(±0.08)   93.61(±0.12)   93.19(±0.17)
> > > CIFAR-100   DenseNet-40     80%     69.60(±0.22)  71.04(±0.36)   71.45(±0.30)   72.01(±0.31)
> > > ========================================================================
> > > CIFAR-10   PreResNet-164   80%     91.76(±0.38)  93.21(±0.17)   93.49(±0.20)    92.14(±0.16)
> > > CIFAR-10   PreResNet-164   90%     82.06(±0.92)  87.55(±0.68)   88.44(±0.19)    85.59(±0.80)
> > > -------------------------------------------------------------------------------------------------------------------------------
> > >
> > > ---------------------------------------------------------------------------------------------------------------------------
> > >   Dataset           Model       Ratio    Fine-tune-restart   Scratch-E-restart  Scratch-B-restart
> > > ---------------------------------------------------------------------------------------------------------------------------
> > > CIFAR-10     DenseNet-40       80%       93.19(±0.17)          93.46(±0.15)         93.23(±0.34)
> > > CIFAR-100   DenseNet-40       80%       72.01(±0.31)          71.71(±0.52)         72.29(±0.41)
> > > =====================================================================
> > > CIFAR-10     PreResNet-164   80%       92.14(±0.16)           93.52(±0.15)        93.15(±0.43)
> > > CIFAR-10     PreResNet-164   90%       85.59(±0.80)           88.07(±0.66)        88.26(±0.45)
> > > ---------------------------------------------------------------------------------------------------------------------------
> > > It can be seen that under this setting, Scratch-E/B with restart outperforms fine-tuning with restart in all cases.

---

> > > ### Author Response · Authors · 2018-12-04
> > > **Response to AnonReviewer1 [1/3]**
> > >
> > >
> > > Thanks for your detailed feedback! We have followed your suggestions in the first review and uploaded a revision, and a summary can be found here (https://openreview.net/forum?id=rJlnB3C5Ym&noteId=SkeqNuj5R7 ). Now we give our response to your new reply and present some new results as follows:
> > >
> > > #1#2. ## Fine-tuning saves training time ## In the newest revision, we have emphasized the fast speed of fine-tuning in both introduction and conclusion. In the 3rd paragraph of intro where we present our main finding, we added "However, in both cases, if a pretrained large model is already available, pruning and fine-tuning from it can still greatly save the training time to obtain the efficient model"; in the conclusion, we make this point more visible by making bullet points on when pruning and fine-tuning is faster; we also mentioned "sometimes there are pretrained models available" in the first paragraph of intro.
> > >
> > > #5. ## Fine-tuning with enough epochs ##
> > > 1) We agree that this experiment shows fine-tuning is faster, and we've emphasized this in the revision. Indeed 320 epochs are longer than the Scratch-B setting in the paper, and here it is for demonstration purpose (showing fine-tuning for more epochs does not bring much improvement compared with scratch-training) and we didn't use the results here in the paper.  Also, if we count both large model training and fine-tuning epochs, scratch-training with 160/320 epochs is still at a  disadvantage compared with fine-tuning with 160/320 epochs, since the latter benefits from the epochs for large model training.
> > >
> > > 2) (Q1) In scratch-training we use the learning rate for large model training (initial learning rate 0.1 multiplied by 0.1 at ½ and ¾ schedule), and for fine-tuning we use a constant low learning rate (0.001). We use this type of learning rate is to follow prior works, and using low learning rate in fine-tuning is also part of the prior belief that "the inherited weights should be preserved". However, we do agree that exploring the learning rate choices of fine-tuning can be useful.
> > >
> > > Here, we investigate using the same learning rate for fine-tuning as the large model training (initial learning rate 0.1 multiplied by 0.1 at ½ and ¾ schedule) and fine-tuning for 160 epochs. We call this "fine-tuning with learning rate restart", since if we include the large model training the learning rate first drops from 0.1 to 0.001 and then after pruning it "restarts" at 0.1 and then again drops to 0.001. We found this can be better than the fine-tuning or scratch-training results in the original paper in the table below (result tables are also available in this pdf link https://drive.google.com/open?id=1dnMDj_kAYblUjHPm9CAGi3bztkyH5dsj ):
> > >
> > > ------------------------------------------------------------------------------------------------------------------------------
> > >   Dataset      Pruned  Model        Fine-tune         Scratch-E        Scratch-B      Fine-tune-restart
> > > -------------------------------------------------------------------------------------------------------------------------------
> > > CIFAR-10       VGG-16-A          93.41(±0.12)    93.62(±0.11)     93.78(±0.15)      93.80(±0.07)
> > > CIFAR-10      ResNet-56-A       92.97(±0.17)    92.96(±0.26)     93.09(±0.14)      93.46(±0.21)
> > > CIFAR-10      ResNet-56-B       92.67(±0.14)    92.54(±0.19)     93.05(±0.18)      93.29(±0.19)
> > > CIFAR-10     ResNet-110-A      93.14(±0.16)    93.25(±0.29)     93.22(±0.22)      93.55(±0.17)
> > > CIFAR-10     ResNet-110-B      92.69(±0.09)    92.89(±0.43)     93.60(±0.25)      93.51(±0.15)
> > > -------------------------------------------------------------------------------------------------------------------------------

---

> > > ### Public Comment · (anonymous) · 2018-12-09
> > > **#5 “fine-tuning with enough epochs”.**
> > >
> > > Dear reviewer,
> > >   I think the discussion above answers your #5 https://openreview.net/forum?id=rJlnB3C5Ym&noteId=H1gtxtsJAQ&noteId=HklijvuWaX
> > >   In short, no matter how long the model is fine-tuned, the comparison is unfair, since the original ImageNet model is not converged. If the original model is converged, pruning methods can achieve better performance.
> > >   The fair setting should be:
> > >   1. train the original model long enough until convergence.
> > >   2. prune the converged model and fine-tune until convergence.
> > >   3. train the pruned model from scratch until convergence.

---

> ### Author Response · Authors · 2018-11-14
> **Response to AnonReviewer1 [3/4]**
>
> 7.  #Fine-tuning is faster# It is true that fine-tuning is faster than training from scratch, and our further explanation is similar to that for point 2.
>
> ## Smaller variance for fine-tuning ##
> The observation about variance is interesting. It seems that this point is most obvious for ResNet-110 in Table 1 (Section 4.1, L1-norm filter pruning), while not so apparent for other methods and models. To investigate whether this is a coincidence or it implies something deeper, we have rerun the experiments for L1-norm pruning on ResNet-110. Here are the results:
> -------------------------------------------------------------------------------------------------
> Pruned Model      Baseline        Fine-tuned       Scratch-E      Scratch-B
> -------------------------------------------------------------------------------------------------
> ResNet-110-A   93.56(±0.19)   93.41(±0.20)   93.06(±0.20)   93.34(±0.21)
> ResNet-110-B   93.56(±0.19)   93.03(±0.21)   92.72(±0.18)   93.64(±0.22)
> -------------------------------------------------------------------------------------------------
> It can be seen that the variance of fine-tuned models’ accuracy is now at the same level with scratch trained models. Thus we think the result of different levels of variance for ResNet-110 in Table 1 might be a coincidence.
>
> The variance result (5 instances) on ImageNet is very expensive to run (can take up to  2 weeks on an 8-GPU machine for one model, previous image classification works [8, 9] also rarely report variance on ImageNet). We will let you know when we have results.
>
> In our experiments, for fine-tuning, the 5 accuracies which we report mean and std, are from 5 different large models, instead of fine-tuning 5 times from the same large model. However, if we fine-tune the *same* pruned model for 5 times, the variance of the final models’ accuracy is indeed smaller (results shown in the table below). The standard deviations are all less than 0.1, in contrast to ~0.2 for those fine-tuned from different large models. This is intuitive given that the same pruned weights are used as initialization and relatively small learning rate is used.
> -----------------------------------------------------------------------------------------------------------------
> Pruned Model       Model-1       Model-2          Model-3         Model-4         Model-5
> ------------------------------------------------------------------------------------------------------------------
> ResNet-110-A   93.10(±0.06)  93.04(±0.03)  92.95(±0.07)  93.48(±0.04)   93.13(±0.07)
> ResNet-110-B   92.85(±0.08)  92.60(±0.05)  92.76(±0.09)  92.98(±0.10)   92.64(±0.07)
> ------------------------------------------------------------------------------------------------------------------
>
> 8. ## Hyperparameters for training/fine-tuning ## We list the hyperparameters used in our experiments below.
>
> On CIFAR, the initial learning rate is 0.1, weight decay is 0.0001 and batch size is 64. We train for 160 epochs and the learning rate is dropped by 0.1 at 80 and 120 epochs. We use SGD with momentum 0.9. Fine-tuning is 40 epochs and using learning rate 0.001, with other settings the same as training.
>
> On ImageNet, the initial learning rate is 0.1, weight decay is 0.0001 and batch size is 256. We train for 90 epochs and the learning rate is dropped by 0.1 at 30 and 60 epochs. We use SGD with momentum 0.9. Fine-tuning uses 20 epochs with learning rate 0.001, with other settings the same as training.
>
> Both settings are very close to the original paper of L1-norm filter pruning, except that we use 160 instead of 164 epochs for training, and use batch size 64 instead of 128. However, the hyperparameter settings (including batch size) are consistent for training the large model, fine-tuning and Scratch-E/B. If smaller batch size leads to better results, it will benefit the large model training and fine-tuning as well, so we believe the comparison is fair. We also have run the experiments using batch size 128 for all training, and the results are below:
> ----------------------------------------------------------------------------------------------------
> Pruned Model       Baseline         Fine-tuned       Scratch-E       Scratch-B
> ----------------------------------------------------------------------------------------------------
> ResNet-56-A      92.26(±0.23)    92.18(±0.34)    92.65(±0.24)   92.63(±0.26)
> ResNet-56-B      92.26(±0.23)    91.82(±0.21)    91.85(±0.22)   92.70(±0.29)
> ----------------------------------------------------------------------------------------------------
> We observe that when the batch size is 128, training from scratch is still at least on par with fine-tuning.

---

> ### Author Response · Authors · 2018-11-14
> **Response to AnonReviewer1 [2/4]**
>
> 5. ## Fine-tuning with enough epochs ## Thank you for asking, this is an important point which was not explained in detail in the paper. Fine-tuning usually uses a small learning rate [1, 2, 4, 5] (usually the final learning rate during large model training) to preserve the inherited weights, which are believed to be helpful for the small model. Using small learning rate for fine-tuning is a part of the previous belief on the necessity of preserving the "important" weights. However, it might cause the model to be stuck in a local minimum. But training pruned model from scratch uses the same learning rate (decays from large to small) as large model training, which is the reason why it can sometimes outperform fine-tuning if enough epochs are trained. We will add this explanation in the revision.
>
> We have done experiments to illustrate the effects of the number of epochs on both fine-tuning and training from scratch. We choose l1-norm pruning on ResNet-56. The results are as follows:
> --------------------------------------------------------------------------------------------------------
> ResNet-56-A       20                   40                 80              160              320
> --------------------------------------------------------------------------------------------------------
> Scratch          86.64(0.41)    90.12(0.26)   91.71(0.13)  92.96(0.26)  93.60(0.21)
> Fine-tune       93.00(0.18)    92.94(0.05)   92.95(0.17)  92.95(0.17)  93.02(0.20)
> --------------------------------------------------------------------------------------------------------
>
> --------------------------------------------------------------------------------------------------------
> ResNet-56-B        20                   40                 80              160              320
> --------------------------------------------------------------------------------------------------------
> Scratch          86.87(0.41)    89.71(0.36)   91.56(0.10)  92.54(0.19)  93.41(0.21)
> Fine-tune       92.66(0.10)    92.67(0.09)   92.68(0.10)  92.64(0.10)  92.70(0.16)
> --------------------------------------------------------------------------------------------------------
> It can be seen that despite fine-tuning is faster to converge, training from scratch is better when the number of epochs is large enough.
>
> 6. ## Significantly pruned models ## The ThiNet model "VGG-Tiny" evaluated in our paper is a significantly pruned model (FLOPs reduced by 15x), and the same observation still holds. We have done more experiments on significantly pruned models using Network Slimming [3]. The pruning ratio for those models is at most 60% in the original paper but here we prune the models by 80% and 90%. Here are the results:
> -------------------------------------------------------------------------------------------------------
>   Dataset           Model             Ratio     Fine-tuned      Scratch-E       Scratch-B
> -------------------------------------------------------------------------------------------------------
> CIFAR-10     DenseNet-40       80%     92.64(±0.12)  93.07(±0.08)   93.61(±0.12)
> CIFAR-100   DenseNet-40       80%     69.60(±0.22)  71.04(±0.36)   71.45(±0.30)
> ===========================================================
> CIFAR-10     PreResNet-164   80%     91.76(±0.38)  93.21(±0.17)   93.49(±0.20)
> CIFAR-10     PreResNet-164   90%     82.06(±0.92)  87.55(±0.68)   88.44(±0.19)
> -------------------------------------------------------------------------------------------------------
> We observe that when the pruned ratio is large enough, training from scratch is better by a even larger margin than fine-tuning.
>
> After the submission, we have run more experiments on non-structured pruning [5] with ImageNet, the results are shown below. We found in some cases training from scratch cannot match the accuracy of fine-tuning.
>
> -----------------------------------------------------------------------------------------------------
>   Dataset         Model           Ratio      Fine-tuned      Scratch-E     Scratch-B
> -----------------------------------------------------------------------------------------------------
> ImageNet      VGG-16         30%           73.68             72.75            74.02
> ImageNet      VGG-16         60%           73.63             71.50            73.42
> ImageNet     ResNet-50      30%           76.06             74.77            75.70
> ImageNet     ResNet-50      60%           76.09             73.69            74.91
> --------------------------------------------------------------------------------------------------
>
> This is possibly due to the fine pruning granularity of non-structured pruning and the task complexity of ImageNet. We further explored the change of weight distribution with non-structured pruning, which could be a reason too. (Refer to Section 1 in this anonymous link for more details, https://drive.google.com/open?id=1BjGJQASV-CuGoq-nVErIRihHMdVwCZxl ) We will include this result and discussion in the revision.

---

> ### Author Response · Authors · 2018-11-14
> **Response to AnonReviewer1 [1/4]**
>
> Thank you for your review and detailed questions! We are happy to address your concerns:
>
> 1. ## Two common beliefs ## The two “common beliefs” indeed can be combined into one statement, but we would like to keep them separate to emphasize two slightly different perspectives: 1) Optimization. Given that a large model presumably provides stronger optimization power, it is believed that training a large model first is necessary for finding optimal "important" weights to condense the model; 2) Initialization. Given "important" weights pruned from a large model, the common belief is that it is necessary to inherit them to achieve a final efficient model, even if we have enough training resources.
>
> 2. ## Training as first step ## We agree that training the large model is not always the first step. In fact, when there exist pretrained models, it is a lot faster to prune and fine-tune than training model from scratch. This point is mentioned at the conclusion part of the paper (second last paragraph). We will further emphasize this point by making it more visible (as one of the benefits of pruning overtraining from scratch, in bullet points at the conclusion) and also state this in our introduction.
>
> However, in many practical applications, pretrained models may not be available, and one has to train specialized large model by him/herself. In addition, some pruning methods (e.g., [1, 2]) impose additional sparsity constraints during the large model training process, in which case one has to train the large model with customized settings and it is not possible to directly get a pretrained model from others.
>
> In the efficient deep learning literature, what seems more important is the *inference speed and model size*, rather than *training time*, because inference/storage sometimes must be on low-end mobile devices while training can be done in high-end GPUs. Most previous works' emphasis is on pursuing a final efficient model for inference on low-resource settings (e.g., see introductions of [5, 6]), rather than optimizing the training time. Further, it was believed that pruning and fine-tuning is not only for fast training speed, but it also gives an efficiency that is not reachable by naive training from scratch (see [1, 2], where training from scratch is reported to be worse than pruning and fine-tuning. We found this is due to a simpler-than-standard data augmentation scheme is used for training from scratch in authors' code and also they didn’t evaluate scratch-B as a baseline). Our work shows that when one is not constrained by training resource and only cares about inference efficiency, pruning from a large model does not give an efficiency (accuracy/resource tradeoff) that is not reachable by direct training from scratch.
>
> 3. ## Time/complexity for pruning/fine-tuning ## Thanks for your suggestion. Here we provide some details about this.
>
> Time: For most pruning methods examined in this paper, pruning takes a negligible amount of time (several seconds). For reconstruction-based methods (regression-based pruning [1] & ThiNet [2]) where pruning is formulated as an optimization problem, it can take longer time (several minutes), but the time is still short compared to training. For fine-tuning, in our experiments, fine-tuning takes at most one-fourth of the standard training schedule. For CIFAR, scratch-training/fine-tuning takes 160/40 epochs. For ImageNet, scratch-training/fine-tuning takes 90/20 epochs. The time for training/fine-tuning is in proportion with the number of epochs for the same model, so the comparison on time is straightforward. We will include this information in the paper.
>
> Complexity: In our experience, for the pruning process, weight-norm based methods are easier to implement, while reconstruction-based methods are not as straightforward. In addition, in training the large model, some special optimization techniques [3, 7] can be required for sparsity regularization, to facilitate the later pruning, which also requires some effort to implement. We have mentioned the engineering effort required as a bullet point in our conclusion, but implementation complexity is a more subjective thing that is hard to precisely measure.
>
> 4. ## Second value of pruning ## Thank you for pointing out. This is a point we are trying to make through some experiments in Section 5 (figure 3 middle and right). In the revision, we will include more results on this point (e.g., more results on network slimming), raise it to a major focus of the paper and mention it in the abstract.

---

### Official Review · AnonReviewer3 · 2018-11-02
**The primary claim is not surprising, but an exciting result is buried at the end**

**Rating:** 6
**Confidence:** 5

**Review:**

This paper proposes to investigate recent popular approaches to pruning networks, which have roots in works by Lecun ‘90, and are mostly rooted in a recent series of papers by Song Han (2015-2016). The methods proposed in these papers consist of the following pipeline: (i) train a neural network, (ii) then prune the weights, typically by trimming the those connections corresponding to weights with lowest magnitude, (iii) fine tune the resulting sparsely-connected neural network.

The authors of the present work assert that traditionally, “each of the three stages is considered as indispensable”. The authors go on to investigate the contribution of each step to the overall pipeline. Among their findings, they report that fine-tuning appears no better than training the resulting pruned network from scratch. The assertion then is that the important aspect of pruning is not that it identifies the “important weights” but rather that it identifies a useful sparse architecture.

One problem here is that the authors may overstate the extent to which previous papers emphasize the fine-tuning, and they may understate the extent to which previous papers emphasize the learning of the architecture. Re-reading Han 2015, it seems clear enough that  the key point is “learning the connections” (it’s right there in the title) and that the “important weights” are a means to achieve this end. Moreover the authors may miss the actual point of fine-tuning. The chief benefit of fine-tuning is that it is faster than training from scratch at each round of retraining, so that even if it achieves the same performance as training from scratch, that’s still a key benefit.

In general, when making claims about other people’s beliefs, the authors need to provide citations. References are not just about credit attribution but also about providing evidence and here that evidence is missing. I’d like to see sweeping statements like “This is
usually reported to be superior to directly training a smaller network from scratch” supported by precise references, perhaps even a quote, to spare the reader some time.

To this reader, the most interesting finding in the paper by far is surprisingly understated in the abstract and introduction, buried at the end of the paper. Here, the authors investigate what are the properties of the resulting sparse architectures that make them useful. They find that by looking at convolutional kernels from pruned architectures, they can obtain for each connection, a probability that a connection is “kept”. Using these probabilities, they can create new sparse architectures that match the sparsity pattern of the pruned architectures, a technique that they call “guided sparsification”. The method yields similar benefits to pruning. Note that while obtaining the sparsity patterns does require running a pruning algorithm in the first place, ***the learned sparsity patterns generalize well across architectures and datasets***. This result is interesting and useful, and to my knowledge novel. I think the authors should go deeper here, investigating the idea on yet more datasets and architectures (ImageNet would be nice). I also think that this result should be given greater emphasis and raised to the level of a major focal point of the paper. With convincing results and some hard-work to reshape the narrative to support this more important finding, I will consider revising my score.

---

> ### Author Response · Authors · 2018-11-27
> **References**
>
>
> [1] Channel Pruning for Accelerating Very Deep Neural Networks. He et al., ICCV 2017.
> [2] ThiNet: A Filter Level Pruning Method for Deep Neural Network Compression. Luo et al., ICCV 2017.
> [3] Pruning Filters for Efficient ConvNets. Li et al., ICLR 2017.
> [4] Learning both Weights and Connections for Efficient Neural Networks. Han et al., NIPS 2015.
> [5] Learning Efficient Convolutional Networks through Network Slimming. Liu et al., ICCV 2017.
> [6] Data-Driven Sparse Structure Selection for Deep Neural Networks. Huang et al., ECCV 2018.
> [7] Deep Compression: Compressing Deep Neural Networks with Pruning, Trained Quantization and Huffman Coding. Han et al., ICLR 2016.
> [8] Pruning Convolutional Neural Networks for Resource Efficient Inference. Molchanov et al., ICLR 2017.
> [9] AutoPruner: An End-to-End Trainable Filter Pruning Method for Efficient Deep Model Inference. Luo et al. arXiv, 2018.
> [10] NISP: Pruning Networks using Neuron Importance Score Propagation. Yu et al., CVPR 2018.
> [11] “Learning-Compression” Algorithms for Neural Net Pruning. Carreira-Perpinan et al., CVPR 2018.

---

> ### Author Response · Authors · 2018-11-27
> **Response to AnonReviewer3 [2/2]**
>
> (continuing on point (3)) In the revision, we have made this point more visible in a bullet point, and in the introduction we've also included "However, in both cases, if a pretrained large model is already available, pruning and fine-tuning from it can still greatly save the training time to obtain the efficient model".
>
> However, in the efficient deep learning literature, researchers usually emphasize the *inference time* and *model size* more than *training time* (e.g., see introductions of [4,7]). The training can be done in high-end clusters while the inference sometimes must be done in low-end mobile devices. Moreover, in practice, pretrained models are not always available like ImageNet models, and in some pruning methods, the pre-training of the large model needs to be customized (e.g., use special sparsity regularization [5,6]) so a normally trained model cannot help. In iterative pruning as in [4], the training time-saving is particularly useful, but later works [1, 2, 3, 6, 9] mostly use one-shot pruning. Combining with point 2, we think the major benefit of keeping the weights was believed to be achieving a good final model.
>
> 4). #Understate Architecture Learning# The prior belief was mentioned as "both the pruned architecture and its associated weights are believed to be essential for obtaining the final efficient model", so we agree that the architecture was believed to be important in previous works too. In the revision, we've emphasized that some prior works (e.g., Han et al. 15 [4]) have made connections between architecture learning and pruning, in the first paragraph of section 5.
>
> However, There are still many works that use predefined pruned architectures [1,2,3], and they didn't mention the learning of architecture in pruning. Also, the popularity of predefined methods supports that, pruning methods are not mostly treated as learning architectures. Indeed, our work is among the first to draw a distinction between predefined methods and automatic methods. Our results also suggest that for predefined pruning methods, one could train the target model directly, which is not previously shown. Even for automatic pruning methods [4,5,6], the emphasis was not mainly on learning architecture: the comparison with uniform pruning or other architecture search methods are not conducted, and the connection with architecture learning is mostly mentioned only in related work.
>
> Others:
> Thank you for pointing out, we have removed the assertion that "each of the three stages is considered as indispensable", as we consider this statement not accurate in describing the previous belief.
>
> 3. ## Experiments and Emphasis on (Transferring) Pruned Sparsity Patterns ##
>
> Thanks for your suggestions on this set of experiments. We have investigated this point on more pruning methods, datasets and architectures. We give a brief summary below, and complete results with analysis can be found in the revision (Section 5 and Appendix F).
>
> Similar to non-structured weight level pruning, we performed the experiments on the channel-level Network Slimming [5]. We introduce two design principles: 1) "Guided Pruning": we use the average number of channels in each layer stage (layers with the same feature map size) from pruned architectures to construct a new set of architectures; 2) "Transferred Guided Pruning": we distill the patterns of pruned architectures to design models on different datasets and architectures. This is similar to "Guided Sparsifying" and "Transferred Guided Sparsifying" for non-structured weight pruning in the original submission.
>
> We present the results of Network Slimming for VGG-19 on CIFAR-100, DenseNet-40 on CIFAR-100 and VGG-19 on ImageNet. In these three cases, we transfer the average pruned sparsity patterns from VGG-16 on CIFAR-10, DenseNet-76 on CIFAR-100 and VGG-11 on ImageNet respectively. We find that architectures obtained by transferred pruned patterns are better than uniformly pruned models, and are close to the pruned architectures, in terms of parameter efficiency. We observe that in these cases, when the prune ratio increases, the later stages are more likely to be pruned. This suggests that there is more redundancy in the later stages and pruning can help us identify it.
>
> In Appendix F, we show that there exist cases when the pruned architectures are not much better than uniformly pruned ones, for both non-structured weight pruning and Network Slimming. In these cases, we find the sparsity patterns in pruned architectures are close to uniform. This might be due to that for those architectures the redundancy is spread more evenly across layers.
>
> Other than presenting more results and analysis in Section 5 and Appendix F, we have also raised this point to a major focal point (e.g., emphasize it more in abstract, introduction, background, etc.) in the revision.
>
> Thank you for your review again! Any further questions or suggestions are welcome.

---

> ### Author Response · Authors · 2018-11-27
> **Response to AnonReviewer3 [1/2]**
>
>
> Thank you for your review! Following your suggestions, we've updated the paper and we're happy to address your concerns. In summary, we would like to explain why we think our finding is surprising, and in the revision we also present more results on the use of sparsity patterns, and raise this point to a major one.
>
> 1. ## References and Quotes about Common Beliefs ##
>
> Our claimed beliefs have references to back up, and we will introduce them in detail here. In [1, 2, 3, 10], pruning and fine-tuning a model is reported to be superior to training the pruned model from scratch. More concretely:
> 1) In Section 4.1.4 and Table 4 of [1], the authors conducted scratch-training experiments and reported that “Shown in Table 4, we observed that it’s difficult for from scratch counterparts to reach competitive accuracy. Our model outperforms from scratch one.”
> 2) In Section 4.2 and Table 1 of [2], the authors compared scratch-training with pruning and reported that “However, if we train this model from scratch, the top-1/top-5 accuracy are only 67.00%/87.45% respectively, which is much worse than our pruned network. ”.
> 3) In Section 4 and Table 1 of [3], the authors showed that “Training a pruned model from scratch performs worse than retraining a pruned model, which may indicate the difficulty of training a network with a small capacity.”.
> 4) In Section 1/4.2 and Figure 3, 4, 6 of [10], the authors compare with scratch-training and reported that “Our experiments show that CNNs pruned by our approach outperform those with the same structures but which are either trained from scratch or randomly pruned.”.
>
> The reason why previous works obtain “contradictory” results with us might be that this baseline is not carefully or properly evaluated.  For example, for [1, 2] we found in authors' code that the accuracy gap could be due to that a simpler-than-standard data augmentation scheme is used in scratch-training. Moreover, previous works didn't compare with Scratch-B.
>
> Here we provide more evidence of the mentioned common beliefs:
> 1) In section 2 of [1], the authors stated that "Many researchers have found that deep models suffer from heavy over-parameterization.... However, this redundancy seems necessary during model training, since the highly non-convex optimization is hard to be solved with current techniques."
> 2) In section 2 of [9], the authors stated that "Pruning is a classic method to reduce network complexity. Compared with training the same structure from scratch, pruning from a pretrained redundant model achieves much better accuracy [2, 3]. This is mainly because of the highly non-convex optimization nature in model training. And, a certain level of redundancy is necessary to guarantee enough capacity during training. Hence, there is a great need to remove such redundancy after training."
> 3) In the first paragraph of [11], the authors stated that “Pruning and compression are possible because these large nets are hugely overparameterized, and empirical evidence suggests it is easier to train a large net and compress it than to train a smaller net from start.”
>
> Thank you for your suggestion, we have added references to the second paragraph of introduction where we introduce previous beliefs in the revision.
>
> 2. ## Why we think our finding is surprising ##
> We would also like to explain why our finding is surprising in four aspects:
>
> 1). #Prior Results# As we mentioned above, previous works have reported that scratch-training cannot match the accuracy of pruning and fine-tuning. In contrast, we revealed that same-level accuracy can be achieved if we ensure a proper and fair scratch-training baseline, thus there's no particular difficulty in training a small model from scratch.
>
> 2). #Necessity of Fine-tuning# In Han et al. 15 [4], the purpose of retaining weights is described as achieving a good final solution instead of saving retraining time: "During retraining, it is better to retain the weights from the initial training phase for the connections that survived pruning than it is to re-initialize the pruned layers. CNNs contain fragile co-adapted features: gradient descent is able to find a good solution when the network is initially trained, but not after re-initializing some layers and retraining them. So when we retrain the pruned layers, we should keep the surviving parameters instead of re-initializing them." Despite the fact that saving time is an important benefit of fine-tuning, it was not brought up as the major one.  We have added this reference in our introduction of "common beliefs".
>
> 3). #Benefit of Fine-tuning# We agree that an important benefit of pruning is to save the training time when given a trained model (as was mentioned in the conclusion section).

---

> > ### Comment · AnonReviewer3 · 2018-12-05
> > **Thanks for the thorough rebuttal and improved draft**
> >
> > I'm satisfied that the author's have addressed many of my primary complaints and improved the exposition of the paper. I have increased my score accordingly.

---

### Official Review · AnonReviewer2 · 2018-11-02
**Interesting results, but not sure they generalize to any pruning approach**

**Rating:** 6
**Confidence:** 4

**Review:**

This paper shows through a set of experiments that the common belief that a large neural network trained, then pruned and fine-tuned performs better than another network that has the same size of the pruned one, but trained from scratch, is actually false. That is, a pruned network does not perform better than a network with the same dimensions but trained from scratch. Also, the authors consider that what is important for good performance is to know how many weights/filters are needed at each layer, while the actual values of the weights do not matter. Then, what happens in a standard large neural network training can be seen as an architecture search, in which the algorithm learns what is the right amount of weights for each layer.

Pros:
- If these results are generally true, then, most of the pruning techniques are not really needed. This is an important result.
- If these results hold, there is no need for training larger models and prune them. Best results can be obtained by training from scratch the right architecture.
- the intuition that the neural network pruning is actually performing architecture search is quite interesting.

Cons:
- It is still difficult to believe that most of the previous work and previous experiments (as in Zhu & Gupta 2018) are faulty.
- Another paper with opposing results is [1]. There the authors have an explicit control experiment in which they evaluate the training of a pruned network with random initialization and obtain worse performance than when pruned and pruned and retrained with the correct initialization.
- Soft pruning techniques as [2] obtain even better results than the original network. These approaches are not considered in the analysis. For instance, in their tab. 1, ResNet-56 pruned 30% obtained a gain of 0.19% while your ResNet-50 pruned 30% obtains a loss of 4.56 from tab. 2. This is a significant difference in performance.

Global evaluation:
In general, the paper is well written and give good insides about pruning techniques. However, considering the vast literature that contradicts this paper results, it is not easy to understand which results to believe. It would be useful to see if the authors can obtain good results without pruning also on the control experiment in [1]. Finally, it seems that the proposed method is worse than soft pruning. In soft pruning, we do not gain in training speed, but if the main objective is performance, it is a very relevant result and makes the claims of the paper weaker.

Additional comments:
- top pag.4: "in practice, we found that increasing the training epochs within a reasonable range is rarely harmful". If you use early stopping results should not be affected by the number of training epochs (if trained until convergence).

[1] The Lottery Ticket Hypothesis: Finding Small, Trainable Neural Networks, Jonathan Frankle, Michael Carbin, arXiv2018
[2] Soft Filter Pruning for Accelerating Deep Convolutional Neural Networks, Yang He, Guoliang Kang, Xuanyi Dong, Yanwei Fu, Yi Yang, arXiv 2018

---

> ### Author Response · Authors · 2018-11-20
> **Reference**
>
> [1] The Lottery Ticket Hypothesis: Finding Small, Trainable Neural Networks, Jonathan Frankle, Michael Carbin, arXiv 2018. https://openreview.net/forum?id=rJl-b3RcF7
> [2] Soft Filter Pruning for Accelerating Deep Convolutional Neural Networks, Yang He, Guoliang Kang, Xuanyi Dong, Yanwei Fu, Yi Yang, arXiv 2018.
> [3] To prune, or not to prune: exploring the efficacy of pruning for model compression. Zhu et al., NIPS workshop 2017.
> [4] Channel Pruning for Accelerating Very Deep Neural Networks. He et al., ICCV 2017.
> [5] ThiNet: A Filter Level Pruning Method for Deep Neural Network Compression. Luo et al., ICCV 2017.
> [6] Learning both Weights and Connections for Efficient Neural Networks. Han et al., NIPS 2015.
> [7] Pruning Filters for Efficient ConvNets. Li et al., ICLR 2017.
> [8] https://github.com/he-y/soft-filter-pruning
> [9] Data-Driven Sparse Structure Selection for Deep Neural Networks. Huang et al., ECCV 2018.
> [10] Learning Efficient Convolutional Networks through Network Slimming. Liu et al., ICCV 2017.

---

> ### Author Response · Authors · 2018-11-20
> **Response to AnonReviewer2 [2/2]**
>
>
> 3. ## Whether our observation holds on soft filter pruning [2] (SFP) ##
>
> First, please note that the two models you mentioned are for two different datasets, i.e., the ResNet-56 is for CIFAR-10, while the ResNet-50 is for ImageNet. These two cases are not directly comparable, since pruning a model for the small CIFAR dataset (50K images, 10 classes) is far easier than compressing a model for the large scale ImageNet task (1.2M images, 1000 classes). Second, in Table 1 of [2], ResNet-56-30% obtaining a gain of 0.19% is a result of running SFP on a *pretrained model*, while our result is not. In Table 1 of SFP [2], for ResNet-56-30%, the result without using pretrained model is 0.49% accuracy drop.
>
> We agree it is meaningful to see whether our observation holds on SFP [2]. We have done experiments to verify our observation on SFP using authors' code [8]. The results are as follows:
>
> CIFAR-10 (not using pretrained models):
> --------------------------------------------------------------------------------------------------------------
>     Model        Ratio Pruned(Paper) Pruned(Rerun)    Scratch-E        Scratch-B
> --------------------------------------------------------------------------------------------------------------
> ResNet-20      10%     92.24(±0.33)     92.00(±0.32)    92.22(±0.15)    92.13(±0.10)
> ResNet-20      20%     91.20(±0.30)     91.50(±0.30)    91.62(±0.12)    91.67(±0.15)
> ResNet-20      30%     90.83(±0.31)     90.78(±0.15)    90.93(±0.10)    91.07(±0.23)
> ===================================================================
> ResNet-32      10%     93.22(±0.09)     93.28(±0.05)    93.42(±0.40)    93.08(±0.13)
> ResNet-32      20%     92.63(±0.37)     92.50(±0.17)    92.68(±0.20)    92.96(±0.11)
> ResNet-32      30%     92.08(±0.08)     92.02(±0.11)    92.37(±0.12)    92.56(±0.06)
> ===================================================================
> ResNet-56      10%     93.89(±0.19)     93.77(±0.07)    93.42(±0.40)     93.98(±0.21)
> ResNet-56      20%     93.47(±0.24)     93.14(±0.42)    93.44(±0.05)     93.71(±0.14)
> ResNet-56      30%     93.10(±0.20)     93.01(±0.09)    93.19(±0.20)     93.57(±0.12)
> ResNet-56      40%     92.26(±0.31)     92.59(±0.14)    92.80(±0.25)     93.07(±0.25)
> ===================================================================
> ResNet-110    10%     93.83(±0.19)     93.60(±0.50)    94.21(±0.39)      94.13(±0.37)
> ResNet-110    20%     93.93(±0.41)     93.63(±0.44)    93.52(±0.18)      94.29(±0.18)
> ResNet-110    30%     93.38(±0.30)     93.26(±0.37)    93.70(±0.16)      93.92(±0.13)
> ---------------------------------------------------------------------------------------------------------------
>
> CIFAR-10 (using pretrained models):
> --------------------------------------------------------------------------------------------------------------
>     Model         Ratio Pruned(Paper) Pruned(rerun)    Scratch-E        Scratch-B
> --------------------------------------------------------------------------------------------------------------
> ResNet-56      30%     93.78(±0.22)     93.51(±0.26)     94.45(±0.30)    93.77(±0.25)
> ResNet-56      40%     93.35(±0.31)     93.10(±0.34)     93.84(±0.16)    93.41(±0.08)
> ResNet-110    30%     93.86(±0.21)     93.46(±0.19)     93.89(±0.17)    94.37(±0.24)
> --------------------------------------------------------------------------------------------------------------
>
> ImageNet (not using pretrained models):
> -------------------------------------------------------------------------------------------
>     Model         Ratio        Pruned       Scratch-E        Scratch-B
> -------------------------------------------------------------------------------------------
> ResNet-34      30%          71.83            71.67                72.97
> ResNet-50      30%          74.61            74.98                75.56
> -------------------------------------------------------------------------------------------
> It can be seen that Scratch-E outperforms pruned models for most of the time and scratch-B outperforms SFP in nearly all cases. Therefore, our observation also holds on SFP.
>
> Additional comments:
> Thank you for your suggestion. In our experiments, we follow the standard training settings for CIFAR and ImageNet, where early stopping is not directly applicable due to the step-wise learning rate schedule.
>
> Thank you for your review again! If you have any further questions, we are very happy to answer.

---

> > ### Comment · AnonReviewer2 · 2018-12-06
> > **More convinced about results**
> >
> > Thanks to the authors for answering to all my questions and doubts. Now I am more convinced about the validity of the obtained results.
> > Regarding the comparison with [1], I checked the last version of the paper. In Fig.8 they show that with their initialization and a special initialization they are able to obtain good performance even with very strong pruning (>80%), while the random initialization obtain lower results. However, also in this case, the reason can be a not sufficient training of the random initialization.
> > Globally, I consider the obtained results very interesting and I think that the paper deserves publication.

---

> > > ### Public Comment · (anonymous) · 2018-12-09
> > > **The scratch-B results are probably not valid**
> > >
> > > Dear reviewer, plz read the discussion above: https://openreview.net/forum?id=rJlnB3C5Ym&noteId=rJlnB3C5Ym&noteId=HklijvuWaX

---

> ### Author Response · Authors · 2018-11-20
> **Response to AnonReviewer2 [1/2]**
>
>
> Thank you for your detailed review! Your questions are valuable, and we are happy to address your concerns. In summary, we explain why our work does *not* contradict with existing works and we show that our observation also holds on the soft filter pruning method [2]. Details are given below:
>
> 1. ##Contradiction with prior works##
> First, we would like to clarify that our results are not contradictory to [3] (Zhu & Gupta). In [3], the authors demonstrate that large sparse models outperform small dense models with the same memory footprint. Here, “large sparse models” refers to the models pruned with non-structured weight level pruning, while “small dense models” refers to models with the same memory footprint to “large sparse models”, but they are of different architectures (one is sparse another is dense). However, in our paper, we compare the same model architectures, and the only difference is that one is obtained by pruning and fine-tuning while the other is trained from scratch. Our results demonstrate that the architecture is more important than inherited weights, while [3] demonstrate a large-sparse architecture is better than a small-dense architecture.
>
> Second, our results are not contradictory to previous works on pruning methods. Previous works either didn't evaluate the scratch-training baseline [2, 6, 9, 10] or didn't choose a strong enough baseline for scratch-training. In [4, 5], while the authors showed that pruned models trained from scratch cannot match the accuracy of pruning and fine-tuning, we found that 1) they used a simpler-than-standard data augmentation scheme for training from scratch in the released code, and 2) they didn’t evaluate under the scratch-B setup.
>
> 2. ## Contradiction with [1] ##
> First, we would like to clarify that our results are not contradictory to results in [1]. The main experiments in [1] are done using non-standard hyperparameters (e.g., very small learning rates) with very small networks, while we use standard hyperparameters with the same modern network architectures as in each pruning method's original paper. In their Appendix D (in the OpenReview version linked below), they show that their hypothesis does not hold when they use standard learning rate on ResNet-18: "We find that, at the learning rate used in the paper that introduced ResNet-18 (He et al., 2016), iterative pruning does not find winning tickets". In their control experiments, that fact that using the "correct initialization" is better could be due to the learning rate being too small, and as a result, the weights are not changed much from the initialization during training. However, the small learning rate they used cannot lead to state-of-the-art performance, which means their hypothesis is more restricted. Moreover, the experiments in [1] are only on non-structured weight pruning, so it is unclear whether the hypothesis holds on other levels of pruning (e.g., channel/filter pruning).
>
> However, we agree that investigating whether using the "correct initialization" could bring benefit in standard training hyperparameters is very useful. We have done the control experiments as in [1] to verify this point. Our results are as follows:
>
> Non-structured weight level pruning:
> ----------------------------------------------------------------------------------------------
>    Dataset            Model          Ratio     "Correct Init"     Random Init
> ----------------------------------------------------------------------------------------------
> CIFAR-10          VGG-19            30%     93.69(±0.13)        93.63(±0.16)
> CIFAR-10          VGG-19            80%     93.58(±0.15)        93.65(±0.19)
> CIFAR-10     PreResNet-110     30%     94.89(±0.14)        94.97(±0.10)
> CIFAR-10     PreResNet-110     80%     93.87(±0.15)        93.79(±0.17)
> CIFAR-100        VGG-19            30%     72.57(±0.58)        72.57(±0.23)
> CIFAR-100        VGG-19            50%     72.75(±0.22)        72.31(±0.19)
> CIFAR-100   PreResNet-110     30%     76.41(±0.15)        76.60(±0.10)
> CIFAR-100   PreResNet-110     50%     75.61(±0.12)        75.48(±0.17)
> -----------------------------------------------------------------------------------------------
>
> L1-norm filter pruning:
> --------------------------------------------------------------------------
> Pruned Model      "Correct Init"          Random Init
> -------------------------------------------------------------------------
> VGG-16-A                93.62(0.09)             93.60(0.15)
> ResNet-56-A           92.72(0.10)             92.75(0.26)
> ResNet-56-B           92.78(0.23)             92.90(0.27)
> ResNet-110-A         93.21(0.09)             93.21(0.21)
> ResNet-110-B         93.15(0.12)             93.37(0.29)
> --------------------------------------------------------------------------
> We observe that when we use standard learning rate and other hyperparameters, using "correct initialization"  does not provide an advantage over random initialization.

---

### Public Comment · ~Yihui_He1 · 2018-10-01
**please use the correct results of channel pruning**

Dear authors,
  I'm Yihui He, the first author of channel pruning and AMC [1] (channel pruning with RL).
  First, please correct our VGG-16 results:

(the baseline top-1/top-5 Err:   29.5%/10.1%)
Model	  Top1/Top5 Inc Err(%)
(Table 1)
VGG_2x_FT		0.0/0.0
VGG_4x_FT		2.2/1.0
VGG_5x_FT		2.9/1.7
(Table 2)
VGG_3C_4x_FT	-0.3/0.0
VGG_3C_5x_FT	0.0/0.3
(Table 4)
From scratch	2.4/1.8
From scratch uni	3.4/2.4
Ours (FT)		2.2/1.0

So actually VGG_2x_FT doesn't hurt accuracy at all. Your argument in section 4.1 "scratch-trained models are better than the fine-tuned models" does not hold for channel pruning. I do agree that channel pruning is not good at ResNet, which is because of the multi-branch characteristic.

Second, I'm more curious to see the comparison of your Scratch-B and Scratch-E with our strong result VGG_3C_4x_FT which is also released on Github. VGG_3C_4x_FT reduced computation 4x and increased top-1 accuracy by 0.3%, which was the state-of-the-art result at the time our paper published.

Third, AMC [1] is a perfect example of network pruning as architecture search (section 5).

[1] AMC: AutoML for Model Compression and Acceleration on Mobile Devices, ECCV'18, http://openaccess.thecvf.com/content_ECCV_2018/html/Yihui_He_AMC_Automated_Model_ECCV_2018_paper.html

---

> ### Author Response · Authors · 2018-10-02
> **Sorry for the typo, "VGG-2x" should be "VGG-5x"**
>
> Dear Yihui,
> Thanks for your detailed comments!
> 1. We double checked our result tables. In our Table 3, we found the "VGG-2x" is actually a typo, it should be "VGG-5x", which is the model available on your GitHub repo and the model we actually used. We are sorry for the confusion and we will correct the typo in the next version. After this change, our results for fine-tuning match the results listed above.
>
> For scratch-trained results, we hypothesize the difference could be explained by less carefully chosen hyper-parameter setting and data augmentation scheme, as we mentioned in Section 1. For example, fine-tuning may not require heavy data augmentation for a good performance (if it is already used in large model training), but training from scratch does require. From your GitHub repo, we observe that you use models pre-trained with heavy data augmentation, and during fine-tuning a simpler data augmentation scheme is used. If the same setting as fine-tuning is used for evaluating the scratch-baseline, the difference could be explained. In our experiment, the scratch-baseline setting is the same as large model training. We plan to release our code soon.
>
> 2. The focus of our submission is the property of network pruning. From our understanding, the "VGG_3C_4x_FT" model you referred to is obtained through a combination of 3 techniques, namely spatial factorization, channel factorization, and channel pruning. If for this model, the scratch results cannot match the fine-tuned results, it could be due to the former two techniques. Thus, to isolate the effects, we choose to evaluate network obtained by only channel pruning.
>
> 3. Yes, thanks for pointing out. The AMC method is indeed very related to our discussion. We will add citations to AMC and other examples of using pruning-related techniques to guide architecture search (e.g., [1]) in the revised version.
> [1] MorphNet: Fast & Simple Resource-Constrained Structure Learning of Deep Networks. CVPR 2018.

---

> > ### Public Comment · ~Yihui_He1 · 2018-10-02
> > **That's interesting**
> >
> > Maybe fine-tuning doesn't rely that much on data augmentation. The training time of the from scratch counterpart in our experiment is not as long as your Scratch-B, which may also cause the difference.

---

> > > ### Author Response · Authors · 2018-10-10
> > > **Data augmentation influence**
> > >
> > > Thank you for the reply. We've conducted experiments to verify the influence of data augmentation (DA) during training and fine-tuning, using the ResNet-34-A/B model from Li et al. Here are the results:
> > > Scratch-E-trained:
> > > -----------------------------------------------------------
> > >      Model          standard DA     simpler DA
> > > -----------------------------------------------------------
> > > ResNet34-A            72.77                70.96
> > > ResNet34-B            72.55                70.89
> > > -----------------------------------------------------------
> > > Fine-tuned:
> > > -----------------------------------------------------------
> > >      Model          standard DA     simpler DA
> > > -----------------------------------------------------------
> > > ResNet34-A            72.56                72.68
> > > ResNet34-B            72.29                71.89
> > > -----------------------------------------------------------
> > > It can be seen that the DA scheme indeed has a much more significant impact on scratch-trained accuracy than fine-tuned accuracy.
> > >
> > > Yes, training for more epochs is another reason for the difference in Scratch-B, which we omitted in our reply since our discussion was on the Scratch-E results.

---

### Public Comment · ~Tijmen_Blankevoort1 · 2018-10-03
**Tensor decompositions**

Interesting results!
Would it be possible to add tensor decomposed architectures like Zhang et. al. or Jaderberg et. al. (Asym3D, Spatial SVD) to the paper? That's a whole set of neural network compression methods left out of the equation here :)

Cheers,
Tijmen

---

> ### Author Response · Authors · 2018-10-04
> **Reply to "Tensor decompositions"**
>
> Thanks for your comment! Indeed, tensor decomposition is a very important family of compression techniques. Here we mainly focus on understanding and verifying the assumptions behind network pruning. Tensor decomposition approaches share some similar operations with network pruning, but differ in some important aspects, e.g., the methods you mentioned (Zhang et al., Jaderberg et al.) do not use fine-tuning, and some works already adopt the strategy of directly training the low-rank decomposed network from scratch (e.g., [1][2]). For tensor decomposition, we think the assumption could be more appropriately described as "low-rank tensor is a more efficient parameterization for convolution weights", rather than "starting with a large model is necessary" or "inheriting important weights is helpful", as we studied in the network pruning methods.
>
> That being said, investigating whether tensor decomposition and other compression methods exhibit similar properties would be an interesting future work.
>
> [1] Training CNNs with Low-rank Filters for Efficient Image Classification. Ioannou et al., ICLR 2016.
> [2] Convolutional Neural Networks with Low-rank Regularization. Tai et al., ICLR 2016.

---

### Public Comment · ~Jian-Hao_Luo1 · 2018-10-08
**Interesting paper!**

Hi,

I am Jian-Hao Luo, the first-author of ThiNet (ICCV'17, TPAMI'18). In fact, I also noticed that pruning is not always better than training from scratch. Sometimes, it may be even worse than training from scratch. But, in general, I think pruning can outperform training from scratch.

I have read your paper, then I will talk about the results shown in Table 2.

1) First, Let us focus on VGG16. In my ICCV paper, the top-1 accuracy of VGG-GAP is 67.34%, 3.69% lower than your baseline accuracy (71.03%). But in Table 2, the accuracy drop is 4.93%. Is there anything wrong? Then, let us translate this table into a more intuitive one:
-------------------------------------------------------------
Strategy    VGG-Conv    VGG-GAP    VGG-Tiny
-------------------------------------------------------------
ThiNet         69.80%        67.34%           58.34%
Scratch-E    68.76%        66.85%           57.15%
Scratch-B    71.71%        68.66%           59.93%
-------------------------------------------------------------
Obviously, ThiNet is better than Scratch-E, i.e., train from scratch in the same epochs. In fact, I also compared with training from scratch in ICCV' paper. Its accuracy is 67% (the same architecture as VGG-Conv, in 80 epochs). The gap between 67% and yours 68.76% may be because of deep learning framework (caffe vs. PyTorch) and image pre-processing.

The next question is about Scratch-B? In my opinion, it is due to more training epoch. VGG-Conv and VGG-GAP save more than 3x FLOPs than VGG16 model. Then, Scratch-B will train 3x epochs than Scratch-E. If you use the official settings of PyTorch, it needs 90 epochs to train VGG16 from scratch. Then I guess you cost 270 epochs to train VGG-Conv and VGG-GAP. But, in my ThiNet pipeline, I only conduct 26 epochs to finish pruning and fine-tuning (see the code in Github[1]). Hence, I think Scratch-B is unfair.

2) Next, let us talk about ResNet-50:
a) The same problem also exists. The reported accuracy of ResNet50-30% is 68.42%, 6.73% lower than your baseline (75.15%). Or do you compare with 76.13%?
b) In my ICCV implementation, the hyper-parameter settings of pruning ResNet50 are not good. Hence, the accuracy drop is obvious. We have improved their performance in the journal version [2]. But they are still lower than your Scratch-E and Scratch-B. Then, I think: (1). ThiNet is not good enough when pruning ResNet models. (2). PyTorch is better than caffe when training ResNet.

Hence, I agree that ThiNet is not good at pruning ResNet. But I do not agree that training from scratch is better than all pruning methods in ResNet. For example, our recent arxiv paper AutoPruner [3] is better than the reported training  from scratch results:
----------------------------------------------------------------
Strategy          ResNet50-30%       ResNet50-50%
----------------------------------------------------------------
AutoPruner          72.53%                  74.22%
Scratch-E              70.92%                  73.31%
Scratch-B              71.57%                  73.90%
----------------------------------------------------------------
My AutoPruner has exactly the same architecture as ThiNet. And it is also conducted within PyTorch. My code is based on the PyTorch official training code [4], and use the same image preprocessing pipelines. Hence, the comparison is fair.

3) Anyway, this is an interesting paper. The observation mentioned in this paper is important for the community. Maybe we should think more about pruning and training from scratch.

-------------------
REFERENCE:
[1]. ThiNet code: https://github.com/Roll920/ThiNet_Code
[2]. ThiNet: Pruning CNN Filters for a Thinner Net. TPAMI, 2018.
[3]. AutoPruner: An End-to-End Trainable Filter Pruning Method for Efficient Deep Model Inference. arxiv, 2018.
[4]. PyTorch ImageNet code: https://github.com/pytorch/examples/blob/master/imagenet/main.py

---

> ### Author Response · Authors · 2018-10-08
> **Thanks and our answers**
>
> Hi Jian-Hao,
> Thanks for your comment! We give our answers to your questions below:
> 1)
> Differences in results:
> The differences between our results and yours are due to the difference in image preprocessing scheme at test time, as we mentioned in the methodology section "For testing on the ImageNet, the image is first resized so that the shorter edge has length 256, and then center-cropped to be of 224×224" (scheme 1).  In your paper, the preprocessing is "images are resized to 256 × 256"  and then "center crop to 224 × 224" (scheme 2), except for the "VGG-Tiny" model which uses scheme 1. We believe scheme 1 is a more commonly used one. During our experiments, we evaluated all models in both schemes, and we show the complete results in the anonymous link here ( https://drive.google.com/open?id=1_nQmJlLGqfDDG7MFyyF3Km0eohdQxQPJ ). The results for scheme 2 should match the results in the original ThiNet paper. The reason why we chose to present results for scheme 1 in the paper is also explained in the linked file. If needed, we could include the results for both schemes in the revised paper or Appendix.
>
> Frameworks:
> In fact, we've reproduced the scratch-trained result in your paper (67%) in Caffe, using the data augmentation and image preprocessing scheme (scheme 2 mentioned above) from your Github, which, however, are different from the training setting of the original VGG trained in Caffe Model Zoo (the model you used as the unpruned model). If we train the original unpruned VGG using this setting in Caffe, we cannot achieve the accuracy of the unpruned VGG reported in your paper either.
> As we mentioned in the paper, the contradiction between our results and yours may be due to a simpler-than-standard data augmentation scheme during training from scratch. We believe the cause is more than the differences in frameworks, since we already compare relative performance drop from unpruned models in each framework.
>
> Scratch-B:
> We only train the ImageNet models for at most 180 epochs (2x90 epochs), as mentioned in the footnote of page 4.
> When a large pretrained model is given, we agree pruning and fine-tuning can be faster, as we mentioned in the last section of the paper.  But in most practical cases, we need to train the large model by ourselves (as popular pretrained models are only on certain datasets like ImageNet), thus we think Scratch-B is a fair setting.
>
> 2)
> a) The reason is the same as that for VGG.
>
> b) The difference between Pytorch and Caffe should not matter that much since we compare relative accuracy drop.
>
> As for AutoPruner, we noticed that there are a pooling layer and a fully connected layer for selecting channels in each convolutional layer, and their parameters can be removed after training. From our understanding, this channel selection module is somewhat similar to the "squeeze-and-excitation" module in the SENet paper [1]. Other than enabling pruning, these modules themselves can give the network stronger representation power and boost the accuracy (as shown in [1]), thus a fair comparison would be training the pruned networks with those modules from scratch, and remove them afterward. However, in our evaluation for training from scratch, ResNet-50% and ResNet-30% are not trained with those modules, since ThiNet does not have those modules. Therefore, we think the results are not directly comparable, despite they may have exactly the same architectures after training. Thus, our results here might not support your claim "pruning can outperform training from scratch".
>
> 3) Thanks for your positive feedback!
>
> [1] Squeeze-and-excitation Networks. Hu et al. CVPR 2018.

---

> > ### Public Comment · ~Jian-Hao_Luo1 · 2018-10-09
> > **image preprocessing scheme**
> >
> > Ok, my suggestion is that if you train a model using scheme 2, you should test it using scheme 2 rather than other preprocessing schemes. Or you can fine-tune it using scheme 1 in several epochs, then the network will recover its accuracy. That is to say, I train a VGG-Conv model using scheme 2, and test its accuracy using scheme 2, then I get 69.80%. But, if I test its accuracy using scheme 1, it may be only 67.80% (just a guess). However, if I fine-tune it using scheme 1 with 1 epoch, I will get 69.80% accuracy again. Hence, I think 69.80% is a more convincing result.

---

> > > ### Author Response · Authors · 2018-10-09
> > > **Reply to "image preprocessing scheme"**
> > >
> > > Yes, we agree that testing preprocessing scheme should be the same as the model's training preprocessing scheme. But the original unpruned VGG in Caffe was trained using scheme 1, however, in the ThiNet paper it is evaluated using scheme 2, thus it is significantly worse (68.36% vs 71.03%), so if we compare relative accuracy drop it is unfair for us.
> > >
> > > Is it ok if we use VGG-Conv/GAP/Tiny evaluated on scheme 2 (69.80% for VGG-Conv), and unpruned models evaluated using scheme 1 (71.03% instead of 68.36% in ThiNet paper)? In that case, every model is tested using the same scheme as its training. We think this is a fair setting. If we reach a consensus, we will make the change for both VGG and ResNet series in the revised version.

---

> > > > ### Public Comment · ~Jian-Hao_Luo1 · 2018-10-09
> > > > **Reply**
> > > >
> > > > Of course, the unpruned models should be 71.03%. This is a wrong setting in my ThiNet paper. I just cite the 68.36% accuracy reported in previous work. After ICCV, I noticed this bug. Hence, the accuracy of unpruned model is changed to 71.50% in the journal version.

---

> > > > > ### Author Response · Authors · 2018-10-10
> > > > > **Updated Result Table for ThiNet**
> > > > >
> > > > > Thanks for your suggestion, we've put the updated result table for ThiNet in the anonymous link here (https://drive.google.com/file/d/1oYuVLkACu4tDBi-wuZDOi0_H6XHfK1NC/view?usp=sharing ), and the table in the paper will be updated in the revised version. Now instead of using the same preprocessing scheme for all models, each model is evaluated using the scheme that it is trained/fine-tuned on.
> > > > >
> > > > > We would like to let other readers know that results for other methods do not have this issue, and the update of ThiNet's results will not affect our main conclusions. This issue is due to that different image preprocessing schemes are used for training and fine-tuning in the original ThiNet paper, and in the original submission, we evaluated all models using the most commonly used scheme.

---

### Author Response · Authors · 2018-10-10
**Code Release**

We have released the code and document to reproduce the results in this anonymous link  (https://drive.google.com/open?id=1HB_1FphsWtbuMAdgHbODSLdAMnj3B6TM ). Links to trained ImageNet models are also included in the document.

---

### Public Comment · ~Yang_He2 · 2018-10-22
**Interesting, but unclear for some points.**

Dear authors,
I am Yang He, the first-author of SFP [1].

In my opinion, the core idea of your paper is about the initialization of the small (pruned) network. This paper claims that "random initialization" of the small model is comparable to, or even better than, the "guided initialization", which is provided by the “three-stage pipeline”, the knowledge of pre-trained network and some selection criterions.

However, the above phenomenon might not be enough to conclude that the “three-stage pipeline is not necessary”, because the compared algorithm [2,4,5,6] in the paper might not reflect the true power of the “three-stage pipeline”.

------------------------------------------------------------------------------------------------------------------

For clarity, we use some notations here:
TSFS - Training small model from scratch (the proposed method).
PIPE1 - “three-stage pipeline” of [2,4,5,6] (not allow the pruned connection to recover when training/fine-tuning).
PIPE2 - “three-stage pipeline” of [1,3] (allow the pruned connection to recover when training/fine-tuning).

Generally, PIPE2 could achieve better performance than PIPE1 because PIPE2 allows the incorrectly pruned connections have a chance to come back [3] and provides a larger optimization space than PIPE1 [1].

However, the proposed conclusion, “three-stage pipeline is not necessary,” is just based on the comparison between TSFE and PIPE1 ([2] for weight pruning and [4,5,6] for filter pruning).
If we only consider PIPE1 (instead of PIPE2), we agree with Jian-Hao Luo that “pruning is not always better than training from scratch”[7]. Because the optimization space provided by TSFS and PIPE1 are similar, it is not surprising that sometimes “random initiation” by TSFS is better (the Table 2,3,5 in the paper) and occasionally “guided initiation” by PIPE1 is better (the Table 1,4,6 in the paper).

------------------------------------------------------------------------------------------------------------------

To draw the proposed conclusion that “three-stage pipeline is not necessary” and to achieve a fairer comparison, we recommend comparing with [3] (for weight pruning) and [1] (for filter pruning) for two reasons.

1. PIPE2 could get a better result than PIPE1.
For example, ResNet-56-A on CIFAR-10 in Table 1, PIPE1[4] achieves -0.17%(“-” means accuracy drop) and TSFS achieves -0.18% when pruning 10.4% FLOPs. While in SFP [1], PIPE2 even achieves +0.30% (“+” means accuracy improvement) when pruning 14.7% FLOPs.

2. In [1,8], the pre-trained knowledge is beneficial for the final performance, which is contradictory to the proposed conclusion.
For example, in [1,8], when pruning 40.8% FLOPs of ResNet-110 on CIFAR10, pruning a pre-trained model achieves +0.18% accuracy, while pruning a scratch model achieves -0.30% accuracy (worse than pruning the pre-trained model).
We believe it is one of the benefits of PIPE2 that takes full advantage of the pre-trained knowledge.


By the way, section 5 is fascinating.

-------------------
REFERENCE:

[1] Soft filter pruning for accelerating deep convolutional neural networks, IJCAI’2018, “https://www.ijcai.org/proceedings/2018/0309”, “https://github.com/he-y/soft-filter-pruning”
[2] Learning both Weights and Connections for Efficient Neural Networks, NIPS’2015
[3] Dynamic Network Surgery for Efficient DNNs, NIPS’2016
[4] Pruning filters for efficient convnets. ICLR’2017
[5] Thinet: A filter level pruning method for deep neural network compression. ICCV’2017
[6] Channel pruning for accelerating very deep neural networks.ICCV’2017
[7] https://openreview.net/forum?id=rJlnB3C5Ym&noteId=rkg1zlP_5m
[8] Extended version of [1], https://arxiv.org/abs/1808.07471v2

---

> ### Author Response · Authors · 2018-10-24
> **Thanks and our response**
>
> Dear Yang,
>
> Thanks for your comment! We give our response as follows:
>
> 1. We cannot agree "the core idea is about the initialization of the small (pruned) network".  Instead, our core idea is to validate whether the common beliefs about network pruning are true. We agree that our experiments are comparing using random and inherited weights to initialize the pruned model, but through this, we are really questioning the existing common beliefs about pruning, like "inheriting weights is useful" or "training a large model first is necessary for obtaining a efficient model", as we have extensively discussed in the paper. Surprisingly our results do not support those beliefs and give us some new understandings, which we think is more important than the choice of initialization for the pruned model in engineering practice.
>
> 2. First, we did not claim that the three-stage pipeline is not necessary for every pruning method. Our results only suggest that for a typical pruning algorithm that fits in our pipeline (PIPE1 in your comment) with predefined architectures, one can skip the pipeline.
>
> The pipeline of our evaluated methods (PIPE1) is the dominantly popular pipeline in the network pruning literature [2,4,5,6,12,13,14,15,16,17,18,19,20], and the PIPE2 procedure you mentioned is adopted much less often [1,3]. From our understanding, the soft filter pruning (SFP) procedure proposed in [1], makes a significant modification to the conventional training process, by dropping out certain channels in every epoch, and this can provide additional regularization ability (as mentioned in [1]). Despite SFP could possibly get better results than PIPE1 methods, it could be due to the regularization effect. What we are interested in this paper, is instead the effect/necessity of over-parameterization. A fair scratch-baseline would be training the pruned model with SFP regularization, instead of training the pruned model normally as in our paper. We had a similar discussion for the AutoPruner method [9] in response to Jian-Hao Luo's comment.
>
> "For example, ResNet-56-A on CIFAR-10 in Table 1, PIPE1 [4] achieves -0.17%(“-” means accuracy drop) and TSFS achieves -0.18% when pruning 10.4% FLOPs. While in SFP [1], PIPE2 even achieves +0.30% (“+” means accuracy improvement) when pruning 14.7% FLOPs."
> Note that the ResNet-56-A model that saves 10.4% FLOPs, and the model you mentioned that saves 14.7% FLOPs, are of *different* architectures, so the TSFS (scratch) result of the first model is *not* comparable to the pruned result of the second model. It is possible that the second pruned model just has a better architecture, and training from scratch can give the same or better performance on the second model than pruning. In our paper, we are only interested in whether training the *same* pruned architecture from scratch can be on par with fine-tuning it, without considering models of similar FLOPs but *different* architectures.
>
> That being said, we agree it is possible that similar conclusions do not hold for methods using PIPE2, as well as other variations of network pruning, e.g., dynamic pruning methods [10][11] (pruning based on current input). For better generality, our experiments are on the most general and widely-used prototype of network pruning (PIPE1), since we cannot exhaustively experiment on all variations of pruning methods. We could add discussions on this point in the revision.
>
> 3. "Because the optimization space provided by TSFS and PIPE1 are similar, it is not surprising that sometimes “random initiation” by TSFS is better (the Table 2,3,5 in the paper) and occasionally “guided initiation” by PIPE1 is better (the Table 1,4,6 in the paper)."
>
> It had been a belief that PIPE1 has stronger optimization power than TSFS (see the first belief mentioned in the 2nd paragraph of paper), that's the reason why people use PIPE1 instead of TSFS even when the target architecture is predefined [4,5,6]. Our experiment results, for the first time, suggest the optimization power is not as different as people used to believe, and we cannot agree that this observation is "not surprising".
>
> 4. "In [1,8], the pre-trained knowledge is beneficial for the final performance, which is contradictory to the proposed conclusion. For example, in [1,8], when pruning 40.8% FLOPs of ResNet-110 on CIFAR10, pruning a pre-trained model achieves +0.18% accuracy, while pruning a scratch model achieves -0.30% accuracy (worse than pruning the pre-trained model)."
> From our understanding, the "scratch" you referred to in your paper, means training a large model from scratch with the SFP, then pruning, and is *different* from the "scratch" we used in our paper, which means training the pruned model from scratch. This result demonstrates SFP is more suitable for fine-tuning a pretrained model, but is not directly related and not contradictory to our conclusion.
>
> 5. Thanks again for your positive feedback on our Section 5, as well as other feedbacks!

---

> > ### Author Response · Authors · 2018-10-24
> > **References for our last response, due to space limit**
> >
> > [9]  AutoPruner: An End-to-End Trainable Filter Pruning Method for Efficient Deep Model Inference. Luo et al. arXiv, 2018.
> > [10] Runtime Neural Pruning. Lin et al. NIPS, 2017.
> > [11] SkipNet: Learning Dynamic Routing in Convolutional Networks. Wang et al. ECCV, 2018.
> > [12] Optimal Brain Damage. Lecun et al. NIPS, 1990.
> > [13] Pruning Convolutional Neural Networks for Resource Efficient Inference. Molchanov et al. ICLR, 2017.
> > [14] Network trimming: A Data-driven Neuron Pruning Approach towards Efficient Deep Architectures. Hu et al. arXiv, 2016.
> > [15] Rethinking the Smaller-norm-less-informative Assumption in Channel Pruning of Convolution Layers. Ye et al. ICLR, 2018.
> > [16] Less is More: Towards Compact CNN. Zhou et al. ECCV, 2016.
> > [17] Data-driven Sparse Structure Selection for Deep Neural Networks. Huang et al. ECCV, 2018.
> > [18] Learning Efficient Convolutional Networks through Network Slimming. Liu et al. ICCV, 2017.
> > [19] Learning Structured Sparsity in Deep Neural Networks. Wen et al. NIPS, 2016.
> > [20] Principal filter analysis for guided network compression. Suau et al. arXiv, 2018.

---

> > > ### Public Comment · ~Yang_He2 · 2018-10-26
> > > **the second round reply from reviewers**
> > >
> > > Thanks for your feedback.
> > > If the paper just focuses on a special type of pruning algorithm (the typical PIPE1), the conclusion makes sense.
> > >
> > > To avoid misunderstanding, it is better to add a proper constraint to the conclusion of the paper. For example:
> > > 1. In the Abstract Section, "For pruning algorithms which assume a predefined architecture of the target pruned network, one can completely get rid of the pipeline and directly train the target network from scratch". PIPE2 may not be included in the range of the “pruning algorithm” in this sentence.
> > > 2. In the Background Section, “Fine-tuning the pruned model with inherited weights is no better than training it from scratch.” Because for PIPE2, utilizing inherited weights is still better than training from scratch.
> > >
> > > For Background, adding some discussions of the variants of the *typical* three-stage pipeline [1,3,8,10,11] is great, we are looking forward to seeing the revision.
> > >
> > > We agree with the authors that “we cannot exhaustively experiment on all variations of pruning methods”, the authors do not need to cut the first two stage of every variation of pruning methods and run the experiment. We think the result comparison is enough and also necessary for two reasons.
> > >
> > > 1. In fact, TSFS, PIPE2 [1,3], and others [10,11] all reconsider PIPE1, but on different aspects. The proposed TSFS consider the initialization of PIPE1 (or the necessity of first two stage of the PIPE1), while PIPE2 is focused on the scheme of training and pruning of PIPE1 (or the necessity of maintaining the network connection of the PIPE1). Therefore, we recommend comparing with those variants to show the importance of the components of the PIPE1, which we believe is important for the community.
> > > 2. [1] achieves a better result than [4,5,6] (maybe state-of-the-art), it would be more convincing if the comparison is revealed in the revision. PIPE2 [1] was adopted much less often than PIPE1 [4,5,6] because PIPE2 [1] is a rather new method.

---

> > > > ### Author Response · Authors · 2018-10-26
> > > > **Authors' Reply**
> > > >
> > > > Thank you for your reply, and we are happy to explain further.
> > > >
> > > > 1. "If the paper just focuses on a special type of pruning algorithm (the typical PIPE1), the conclusion makes sense."
> > > >
> > > > The phrase "special type" is misleading: this pipeline is of dominant popularity in the network pruning literature, and is the most general one, as we have already mentioned in our last response.
> > > >
> > > > 2. "To avoid misunderstanding, it is better to add a proper constraint to the conclusion of the paper. For example: 1. In the Abstract Section, 'For pruning algorithms which assume a predefined architecture of the target pruned network, one can completely get rid of the pipeline and directly train the target network from scratch'. PIPE2 may not be included in the range of the “pruning algorithm” in this sentence."
> > > >
> > > > We don't think the mentioned sentence in the abstract will cause misunderstanding. Here the "pipeline" clearly refers to the typical pipeline described in the second sentence of our abstract. SFP does not fall into this, since in SFP pruning happens with training.
> > > >
> > > > 3. "In the Background Section, 'Fine-tuning the pruned model with inherited weights is no better than training it from scratch.' Because for PIPE2, utilizing inherited weights is still better than training from scratch."
> > > >
> > > > a) We stated in our last response that our "training from scratch" is different from the "training from scratch" in your SFP paper: ours means "training pruned model from scratch", yours means "training large model with SFP from scratch". b) In your paper, utilizing the pretrained weights can be helpful, but at the same time it consumes more computation budget (pretraining + fine-tuning), thus the "training from scratch" baseline in your paper should be trained for more epochs for a fair comparison. The advantage of "utilizing pretrained weights" over "training from scratch" (in your paper) could be merely due to more epochs are trained or more computation budget is used.  c) This statement is for the typical pipeline, not for SFP, as well as all other statements/conclusions in our paper. Considering a), b) and c), we think your result is not against our statement.
> > > >
> > > > 4. We will include references to the mentioned variations of pruning, and we agree that investigating whether similar conclusions hold on SFP is interesting future work. Thanks for your suggestion.

---

### Public Comment · ~Ting-Wu_Chin1 · 2018-10-24
**Interesting results but does this hold for compact model?**

Dear authors,

This work is really interesting. I'm wondering if the same conclusion holds for a much more compact network such as MobileNet or SqueezeNet. I'm thinking along this line since I have done some experiments previously of MobileNetV2 on CIFAR-10 and I found that training from scratch is worse than training from the pruned model by more than 1% with the same architecture and longer time (200 epochs vs 60 epochs). I find this interesting since I do not observe this for ResNet-56 on CIFAR-10 (training from scratch == training from the pruned model).
Specifically, for MobileNetV2 trained from scratch for 200 epochs (same hyperparam with training from scratch), I get accuracy 91.77 with std 0.13. On the other hand, fine-tuning from the pruned model for 60 epochs, I get accuracy 93.07 with std 0.17.

Also, I would like to provide another reference, LcP [1] (my recent work), which can be also recognized as pruning as an architecture search.

[1] Chin, Ting-Wu, Cha Zhang, and Diana Marculescu. "Layer-compensated Pruning for Resource-constrained Convolutional Neural Networks." arXiv preprint arXiv:1810.00518 (2018).

Thanks,
Ting-Wu

---

> ### Author Response · Authors · 2018-10-25
> **The DenseNets we evaluated are more compact than MobileNets**
>
> Dear Ting-Wu,
>
> Thanks for your comment and question! It is really a important point and we will explain below.
>
> The DenseNet-40, and DenseNet-BC-100 we evaluated in our experiments, are actually more compact than MobileNet. It can be seen from the table below. The results are all trained by 160 epochs using standard hyperparameters.
>
> --------------------------------------------------------------------
>      Model          Accuracy (CIFAR-10)    Parameters
> --------------------------------------------------------------------
> DenseNet-40           94.10±0.12                 1.0M
> DenseNet-BC-100   95.24±0.17                 0.8M
> MobileNet_v2          93.67±0.10                 1.1M
> --------------------------------------------------------------------
> (MoibleNet_v2 is adopted from [1])
> Our observations on these DenseNets are consistent with other bigger networks, so it can be argued that our observation hold on relatively compact models. We agree that it would be helpful to include results on MobileNet, and we will let you know when we have results.
>
> The reason why you got worse results on pruned MobileNet when training from scratch, may be that you use Scratch-E rather than Scratch-B. In our experiments, we found using Scratch-B is rather important for extremely small or aggressively pruned models, since it costs significantly less computation than training the large model for the same epochs. This can be seen from our discussion on the VGG-Tiny model in the paper:
> "The only exception is Scratch-E for VGG-Tiny, where the model is pruned very aggressively from VGG-16 (FLOPs reduced by 15×), and as a result, drastically reducing the training budget for Scratch-E. The training budget of Scratch-B for this model is also 7 times smaller than the original large model, yet it can achieve the same level of accuracy as the fine-tuned model."
>
> Also, we suggest not to compare the scratch epochs to fine-tuning epochs directly (200 vs 60 in your comment), as fine-tuning is based on a pretrained large model which is trained possibly using more computation budget.
>
> Thanks for providing your reference, we will include it into our discussion in the revision.
>
> [1] https://github.com/kuangliu/pytorch-cifar/blob/master/models/mobilenetv2.py

---

### Public Comment · ~Brendan_Duke1 · 2018-11-01
**Section 5 W.r.t. Comparison with ENAS, DARTS**

Hi, thanks for the interesting investigation, and releasing the code.

I believe there might be a slight mistake in the last paragraph of Section 5, where it says "it would be interesting to investigate whether training from scratch would sometimes yield better results". Actually, (Pham et al., 2018) and (Liu et al., 2018b) do retrain their discovered architectures from scratch, and both papers say this is important for them to achieve their published results.

Edit: Sorry, I realized that this paragraph seems to appear only in the arxiv version of your paper.

---

> ### Author Response · Authors · 2018-11-02
> **Thanks**
>
> Hi Brendan,
>
> Thanks for your comment! The "sharing/inheriting weights", and "investigate whether training from scratch would sometimes yield better results" we mentioned, are for the training during the search process (for accelerating convergence), not for training the final discovered model. Thanks for bringing this into our attention, we will try to make this more clear in the revision.

---

### Public Comment · (anonymous) · 2018-11-07
**Doubt on the main argument "training an over-parameterized model is not necessary to obtain an efficient final model"**

Dear authors,

There is a theoretical paper named "On the Optimization of Deep Networks: Implicit Acceleration by Overparameterization" in ICML2018 have proved that over-parameterization can accelerate the convergence of a deep neural network, which is just in contrast to your main claim.

I think it is not appropriate to give such a claim from several small experiments because these results may largely depend on the hyper-parameters you set.

---

> ### Comment · AnonReviewer3 · 2018-11-07
> **Inappropriate**
>
> This comment is inappropriate and an abuse of the anonymous commenting system. First, do not try to strengthen your point with 4 exclamation marks. This is a peer reviewing institution, not Reddit. Second, "such a sloppy claim" is a not a professional way to speak to your colleagues.
>
> Third, ~"there is a theoretical paper at ICML that claims something which superficially seems to disagree with you" does not provide clear evidence as to who if anyone is wrong or in what way.
>
> If you would like to edit your comment to provide a more thoughtful analysis of both claims and how you would like to see the paper revised, please do. Else I will ask the Area Chair to purge this comment.

---

> > ### Public Comment · (anonymous) · 2018-11-07
> > **Response**
> >
> > Hi,
> > I have revised the comments and I think my question is fair and reasonable enough now.

---

> > > ### Comment · AnonReviewer3 · 2018-11-07
> > > **Better**
> > >
> > > The edited comment is more suitable. Now that we can put issues of propriety aside, it's not clear to me that the Arora paper is as contradictory as you claim. 1) Their theoretical analysis is based on a large number of assumptions that do not hold here, including *linear* neural networks. 2) They are not analyzing sparse networks with learned connectivity patterns as these authors are.
> > >
> > > Correct me if you're wrong, but I believe that the perceived contradiction here owes to a shallow reading of both papers.

---

> > > ### Author Response · Authors · 2018-11-07
> > > **Different aspects of over-parameterization**
> > >
> > > Thank you for your question, and we appreciate Reviewer 3's effort for his prompt and correct explanations. As Reviewer 3 pointed out, the assumption in [1] is that the neural network only consists of linear layers. More importantly, further explaining his/her second point, [1]'s focus is on over-parameterization's effect on accelerating convergence, while our focus is on its necessity for obtaining a final efficient model (provided that we already know the architecture of the final model), i.e., whether we need to train an over-parameterized model first to obtain a final efficient model. They are from different aspects, so the conclusions are not contradictory.
> > >
> > > As another point, we conducted extensive experiments (multiple pruning methods, datasets, models, pruning ratios, tasks) using standard hyperparameters (same with the standards in the image classification literature [2, 3] and the hyperparameters used in original papers of these pruning methods), so it might be unfair to say that the conclusions are from "several small experiments", and the conclusions are inappropriate "because these results may largely depend on the hyper-parameters you set."
> > >
> > > [1] On the Optimization of Deep Networks: Implicit Acceleration by Overparameterization. ICML 2018.
> > > [2] Deep Residual Learning for Image Recognition. CVPR 2016.
> > > [3] Densely Connected Convolutional Networks. CVPR 2017.

---

### Public Comment · ~Yihui_He1 · 2018-11-08
**conclusions drawn from Scratch-B need rethinking**

I conducted an experiment on VGG-16 ImageNet to verify my point https://openreview.net/forum?id=rJlnB3C5Ym&noteId=Bylhl2NecQ&noteId=Bylhl2NecQ
Longer training schedule matters, but not training from scratch.

An unpruned VGG-16 was trained from scratch with 2x schedule using the authors' code. So far it only finished 132 epoches, but the accuracy already reached 74.5% (v.s. 71.5% 1x schedule).
(The 2x schedule follows Scratch-B, namely learning rate decay happens at 60, 120 epoches. 1x schedule is the normal 90 epoches training schedule.)

Let's put the correct accuracy in Table 3:
                      schedule       unpruned   VGG-16-5x
Fine-tuned     1x                    71.03         −2.67
Scratch-E        1x                    71.51         −3.46
Scratch-B        2x                   74.5            −3.51

We can observed that:
1. scratch-B mainly benefits from longer training schedule.
2. the accuracy drop is larger than Fine-tuned method.
Therefore "scratch-trained models are better than the fine-tuned models" does not hold for channel pruning.
Similarly in Table 2, Scratch-B might be worse than ThiNet.

I also planned to conduct 2x schedule experiments on ResNet, unfortunately I don't have the computational resources for that.

---

> ### Author Response · Authors · 2018-11-08
> **Unfair comparison between large model 2x schedule and current Scratch-B results**
>
> Thank for your experiment results. However, "Scratch-B" means training the pruned model using the same computation budget as training the large unpruned model. If you extend the epochs for training the large unpruned model, the epochs for Scratch-B should also be extended, for it to still be "Scratch-B(udget)". Otherwise, the budgets are not equal any longer. Moreover, for a fair comparison, the fine-tuning result also needs to be based on this new unpruned model which is trained longer. Thus the results are not directly comparable here.
>
> For some pruning methods, we've tried to extend the epochs for both training the large unpruned model and Scratch-E/B, and we found our observations still hold (Scratch-B can match the accuracy of fine-tuning from a large model). But to keep our results comparable with existing literature, we use the standard training epochs for the large model, based on which we determine the epochs for Scratch-E/B, for the experiment results presented in our paper.

---

> > ### Public Comment · ~Yihui_He1 · 2018-11-08
> > **My point is not about computational budget**
> >
> > Thanks for the reply.
> >  The point is that some ImageNet models are not sufficiently trained. That’s why scratch-B with longer training schedule performs better.
> >   Furthermore, pruning methods try to approximate the original model instead of optimizing the final accuracy. It’s unfair for pruning methods to approximate a worse model when compared with scratch-B. So 2x model should be used for pruning for fair comparison with scratch-B.
> >   Therefore it is hard to believe that scratched models are better with experiments of scratch-E.
> >
> > For the second paragraph, please show your detailed experiments.

---

> > > ### Author Response · Authors · 2018-11-08
> > > **Scratch-B is a valid baseline**
> > >
> > > Thanks for your explanation. More epochs of Scratch-B come from the fact that pruned models take less budget to train for each epoch as we mentioned in the paper. If a 2x schedule large model is used, scratch-B should be further extended to ensure a fair comparison. In our opinion, Scratch-B is a valid baseline, especially for predefined pruning methods.
> > >
> > > The experiment mentioned in the last reply is a sanity check and we didn't record the full results in precise number, so it needs rerunning. Currently, in the rebuttal period, we're giving higher priority to experiments which address official reviewers' concerns, and we will let you know the results when we have the resource to run that experiment.

---

> > > > ### Public Comment · ~Yihui_He1 · 2018-11-09
> > > > **Scratch-B is not a valid baseline**
> > > >
> > > > "the fact that pruned models take less budget to train for each epoch." is understood.
> > > > But you still ignore the important fact that Scratch-B is better because of better convergence.
> > > > How can it be a valid baseline when other models you compared didn't even converge?

---

> > > > > ### Author Response · Authors · 2018-11-09
> > > > > **Disagree; will not further reply to avoid distraction**
> > > > >
> > > > > We have stated that if we extend both epochs of large model training and Scratch-B to be more than the standard number of epochs, the same observations still hold. Further, the same number of epochs in image classification literature [1, 2] and your own paper [3] are used in our experiment, for training the large model. If the number of epochs is not enough for "converging", those papers should be corrected instead of ours.
> > > > >
> > > > > The commenter also seems to misuse the comment function of OpenReview: the topic is from a discussion in a previous thread (https://openreview.net/forum?id=rJlnB3C5Ym&noteId=Bylhl2NecQ&noteId=Bylhl2NecQ , mentioned at his start), so a new thread should not be opened to avoid distracting other readers from official reviews. To avoid further distraction, we will not reply to this thread anymore.
> > > > >
> > > > > [1] Deep Residual Learning for Image Recognition. He et al., CVPR 2016.
> > > > > [2] Densely Connected Convolutional Networks. Huang et al., CVPR 2017.
> > > > > [3] Channel Pruning for Accelerating Very Deep Neural Networks. He et al., ICCV 2017.

---

> > > > > > ### Public Comment · ~Yihui_He1 · 2018-11-09
> > > > > > **It would be better to add those sanity check results before drawing conclusions**
> > > > > >
> > > > > > 1. If those sanity check results are not recorded, is it still safe to draw the conclusions from the existing correct experiments? https://openreview.net/forum?id=rJlnB3C5Ym&noteId=Bklxok8M6m&noteId=HklijvuWaX
> > > > > > 2. "Some ImageNet models are not sufficiently trained" is a common sense. I don't think the argument: "those papers should be corrected instead of ours" is solid.
> > > > > > 3. I believe this is a new topic. The previous topic is about your incorrect results of channel pruning.

---

> > > > > > > ### Author Response · Authors · 2018-11-18
> > > > > > > **Sanity check results**
> > > > > > >
> > > > > > > As we mentioned before, when the large model VGG is trained for 180 epochs, the pruned model VGG-5x should be trained for 360 epochs for scratch-B (actually it should be 900 epochs since the model saves 5x Flops, but we use 360 here). Now we have the result for this case:
> > > > > > > ----------------------------------------------------------------
> > > > > > >                               unpruned          VGG-16-5x
> > > > > > > ----------------------------------------------------------------
> > > > > > >  Original paper     71.03          −2.67 (fine-tuned)
> > > > > > >  Ours                      74.78          −2.55 (scratch-B)
> > > > > > > ----------------------------------------------------------------
> > > > > > > We can observe that the accuracy drop is smaller than fine-tuned method. Therefore our observation still holds.
> > > > > > >
> > > > > > > For L1-norm filter pruning, we have also done experiments to extend the large model training schedule from 160 to 300 epochs. The results are as follows:
> > > > > > > --------------------------------------------------------------------------------------------------------
> > > > > > > Pruned Model       Baseline         Fine-tuned         Scratch-E          Scratch-B
> > > > > > > --------------------------------------------------------------------------------------------------------
> > > > > > > ResNet-110-A     93.82(±0.32)     93.75(±0.24)     93.80(±0.15)      94.10(±0.12)
> > > > > > > ResNet-110-B     93.82(±0.32)     93.36(±0.28)     93.75(±0.16)      93.90(±0.17)
> > > > > > > --------------------------------------------------------------------------------------------------------
> > > > > > > It can be seen that scratch trained models still consistently outperforms fine-tuned models.
> > > > > > >
> > > > > > > We are considering including these results in Appendix.

---

> > > > > > > > ### Public Comment · ~Yihui_He1 · 2018-11-18
> > > > > > > > **The first result simply means VGG-16 still need more epochs to converge.**
> > > > > > > >
> > > > > > > > (1) Since VGG-5x with 360 epochs is better than VGG-5x with 180 epochs, you should check the convergence of VGG with 360 epochs.
> > > > > > > >
> > > > > > > > (2) I don't think experiments with L1-norm filter pruning are convincing, because L1-norm filter pruning is the first and the worst channel pruning method.

---

> > > > > > > > > ### Author Response · Authors · 2018-11-19
> > > > > > > > > **Reply**
> > > > > > > > >
> > > > > > > > > In practice, it is not possible to train the model for infinitely long. We've already extended the large model training epochs to 2x standard, and the Scratch-B for VGG-5x actually uses 2.5x less training budget so the fine-tuning result is already at a significant advantage. We think this experiment is enough to support our point. Also, we don't think being the first channel pruning method makes L1-norm pruning the "worst" or its results "not convincing".

---

> > > > > > > > > > ### Public Comment · ~Yihui_He1 · 2018-11-19
> > > > > > > > > > **reply**
> > > > > > > > > >
> > > > > > > > > > The point is that the performance is gained from convergence.
> > > > > > > > > > If VGG with 360 epochs converges but not VGG-5x 360 epochs, then you train VGG-5x for 720 epochs and get better results,  that experiment will be convincing but not this one. Maybe you can try smaller models like MobileNets. They take less time to train.
> > > > > > > > > >
> > > > > > > > > > It is a fact that L1-norm pruning is the worst among the channel pruning methods listed in your experiments.

---

> > > > > > > > > > > ### Author Response · Authors · 2018-11-19
> > > > > > > > > > > **Reply**
> > > > > > > > > > >
> > > > > > > > > > > We understand your point about convergence. But if the "convergence" you mentioned takes an inaffordable budget/unreasonable epochs to achieve, it is not meaningful to consider in practice. In practice, one does not try to achieve every marginal benefit by training the model for unreasonably long. Also, convergence are not only measurable in epochs; if we consider computations, scratch-B and large model are trained to the same degree of convergence.
> > > > > > > > > > >
> > > > > > > > > > > "It is a fact that L1-norm pruning is the worst among the channel pruning methods listed in your experiments." This statement needs to be backed up by a direct comparison on the same model/pruned with other methods. And even if it is slightly worse than other methods, it does not make the results "not convincing".

---

> > > > > > > > > > > > ### Public Comment · ~Yihui_He1 · 2018-11-19
> > > > > > > > > > > > **comparison under the same computations does not make sense**
> > > > > > > > > > > >
> > > > > > > > > > > > 1) Commonly convergence is measured by epochs. Squeezenet is hundreds times smaller than vgg. Why not train it with 1 epoch?
> > > > > > > > > > > > More importantly, people care about inference time of compact models but not training time. It requires weeks to train vgg several years ago, but now it only requires days.
> > > > > > > > > > > >
> > > > > > > > > > > > 2) Comparing under the same training time budget is not reasonable. If training time budget is the concern, why not use pruning-finetuning?
> > > > > > > > > > > > Pruning-finetuning only requires roughly 9 epochs, since it starts from the pre-trained models. Then according to the definition of scratch-B(udget), scratch-B should only be trained for 18 (9x2) epochs for fair comparison.
> > > > > > > > > > > >
> > > > > > > > > > > > 3) If the main point of your paper is ImageNet models are not converged, I would totally agree.
> > > > > > > > > > > > For example:
> > > > > > > > > > > > (1) Suppose the original model is trained for 10 iterations and gets 1% acc.
> > > > > > > > > > > > (2) Then the model is pruned and finetuned for 1 iteration. Suppose it recovers the acc to 1%.
> > > > > > > > > > > > (3) Scratch-B is trained 20 for iterations and gets 5% acc.
> > > > > > > > > > > > Can the conclusion "scratch-trained models are better than the fine-tuned models" be drawn from this experiment?

---

> > > ### Public Comment · (anonymous) · 2018-11-10
> > > **support your argument**
> > >
> > > I also support your opinion that a main problem is that "some models are not sufficiently trained",  as a result it is not sufficient to reach the conclusion.

---

### Author Response · Authors · 2018-11-27
**Summary of Revision**


Thanks for all the detailed reviews! Following reviewers' suggestions, we have updated the paper and uploaded a revision (Nov 26), and here we give a summary of the major changes.

In response to Reviewer 1:
1. We include the fine-tuning details in Section 3 and the results and analysis on fine-tuning for more epochs in Appendix D.
2. We add the results for significantly pruned models in Appendix C.
3. We add more results for non-structured weight pruning on ImageNet in Table 6, with analysis on why in some cases training from scratch cannot match fine-tuning in Section 4.2 and Appendix G.
4. We emphasize the fast speed of fine-tuning (intro and conclusion) and prior works' discussion on pruning and architecture learning (first paragraph of Section 5).

In response to Reviewer 2:
1. We show our observations also hold on soft filter pruning [1] in Appendix A.
2. We present some experiments and analysis on the lottery ticket hypothesis [2] in Appendix B.

In response to Reviewer 3:
1. We add more references when introducing the common belief in the introduction.
2. We include more experiments and analysis on (transferring) pruned sparsity patterns in Section 5 and Appendix F. We also raise this point to a major focus of the paper (e.g., in abstract, intro and conclusion).
3. We emphasize the fast speed of fine-tuning (intro and conclusion) and prior works' discussion on pruning and architecture learning (first paragraph of Section 5).

Others:
1. We add experiments on extending the training epochs in Appendix E.
2. We visualize the weight distribution in Appendix G.
3. We discuss pruning and conventional architecture search methods in the last two paragraphs of Section 5.
4. We update the result table of ThiNet (Table 2) following the suggestion from the original authors.
5. We add related references and discussions as suggested by other commenters.

[1] Soft Filter Pruning for Accelerating Deep Convolutional Neural Networks, Yang He, Guoliang Kang, Xuanyi Dong, Yanwei Fu, Yi Yang, IJCAI 2018.
[2] The Lottery Ticket Hypothesis: Finding Small, Trainable Neural Networks. Anonymous, Submitted to ICLR, https://openreview.net/forum?id=rJl-b3RcF7 .

---

### Author Response · Authors · 2019-03-05
**Camera-ready version**

We uploaded a camera-ready version of the paper. In response to AC's comment, we added more results comparing with the Lottery Ticket Hypothesis in Appendix A, and changed the terminology of "standard" and "non-standard". We would like to thank the AC for the valuable suggestion.

---

### Meta-Review · Area_Chair1 · 2018-12-17
**Empirical paper casting shade on pruning**

**Confidence:** 3
**Recommendation:** Accept (Poster)

**Metareview:**

The paper presents a lot of empirical evidence that fine tuning pruned networks is inferior to training them from scratch. These results seem unsurprising in retrospect, but hindsight is 20-20.  The reviewers raised a wide range of issues, some of which were addressed and some which were not. I recommend to the authors that they make sure that any claims they draw from their experiments are sufficiently prescribed. E.g., the lottery ticket experiments done by Anonymous in response to this paper show that the random initialization does poorer than restarting with the initial weights (other than in resnet, though this seems possibly due to the learning rate). There is something different in their setting, and so your claims should be properly circumscribed. I don't think the "standard" versus "nonstandard" terminology is appropriate until the actual boundary between these two behaviors is identified. I would recommend the authors make guarded claims here.